# Formal Models of Active Learning from Contrastive Examples

**Farnam Mansouri**
University of Waterloo
f5mansou@uwaterloo.ca

**Hans U. Simon**
Ruhr-University Bochum
hans.simon@rub.de

**Adish Singla**
Max Planck Institute for Software Systems
adishs@mpi-sws.org

**Yuxin Chen**
University of Chicago
chenyuxin@uchicago.edu

**Sandra Zilles**
University of Regina and Amii
sandra.zilles@uregina.ca

## Abstract

Machine learning can greatly benefit from providing learning algorithms with pairs of contrastive training examples—typically pairs of instances that differ only slightly, yet have different class labels. Intuitively, the difference in the instances helps explain the difference in the class labels. This paper proposes a theoretical framework in which the effect of various types of contrastive examples on active learners is studied formally. The focus is on the sample complexity of learning concept classes and how it is influenced by the choice of contrastive examples. We illustrate our results with geometric concept classes and classes of Boolean functions. Interestingly, we reveal a connection between learning from contrastive examples and the classical model of self-directed learning.

## 1 Introduction

In machine learning, contrastive data has been used for various purposes, most notably in order to reduce the number of training samples needed to make high-quality predictions, or to explain the predictions of black-box models. A contrastive example for a labeled training data point $(x, b)$ with label $b \in \{0, 1\}$ could be, e.g., a labeled point $(x', 1 - b)$, such that $x'$ fulfills some additional predefined property. This additional property could be defined in a way so as to convey helpful information for learning. For instance, a constraint on $x'$ could be that it is the *closest* point to $x$ (in the given pool of data points, and with respect to a fixed underlying metric) that has a different label from $b$. Such examples are often called counterfactuals and have proven useful in various learning settings, including explanation-based supervised learning [8], but also reinforcement learning [16, 22], and learning recommender systems [24].

Intuitively, counterfactuals present a learning algorithm with an important feature or property of a data object that crucially affects its classification. For example, a training data point $x$ might be an image of a cat, together with its class label (the subspecies of cat). A counterfactual piece of information might be an image $x'$ of a cat that looks similar to the cat in $x$, yet belongs to a different subspecies. The main feature distinguishing $x'$ from $x$ could be highlighted in the image, as a means of explaining the classification of $x$. Similarly, contrastive information can be used in natural language processing and many other applications, and was applied successfully in representation learning, see,

39th Conference on Neural Information Processing Systems (NeurIPS 2025).

for example, [5, 7, 19, 20]. Interestingly, also human learners exhibit improved classification abilities when they are taught with the help of such visual explanations [21].

Theoretical studies have analyzed the sample complexity in specific models of learning with contrastive examples, such as PAC-learning of distance functions [3], or active learning with a greedy teacher [26], a randomized teacher [23] or an adversarial teacher [9]. Other formal studies focus on specific aspects of learning with contrastive information, for instance, how negative examples influence the learning process [5] or how individual loss functions affect contrastive learning [15].

One example of the use of contrastive information, particularly relevant to our study, is counterexample-guided program synthesis (CEGIS) in formal methods research [2]. Consider the task of synthesizing a program that satisfies a given specification $S$. In CEGIS, a synthesizer generates a candidate program $P$, which is then checked against the specification $S$ by a verifier. If $P$ violates $S$, the verifier returns a counterexample witnessing the violation. This process is iterated until a program satisfying $S$ is obtained. We can see each candidate program $P$ as a query to the verifier, which acts as an oracle. Importantly, in this setting, the oracle follows a *known procedure*. In particular, one can implement the oracle to return a specific counterexample (e.g., a smallest one) that acts as a contrastive example, and the synthesizer can be *tailored to the knowledge of how the counterexample is generated*. For example, if the synthesizer knows that a smallest counterexample is generated, it can rule out all programs for which there would have been smaller counterexamples.

Motivated by such applications in program synthesis, the main goal of this paper is to introduce and analyze formal settings of learning from contrastive examples, specifically modeling situations in which the learner has *knowledge of the way in which the oracle selects contrastive examples*.

We propose a generic formal framework, encompassing various ways in which contrastive examples are chosen (i.e., not just as counterfactuals); this is the first contribution of our paper. We then analyze several settings in this framework w.r.t. the resulting sample complexity, i.e., the number of examples required for identifying the underlying target concept, resulting in various non-trivial upper and lower bounds on learning classes of geometric objects or Boolean functions (most notably 1-decision lists and monotone DNFs); this constitutes our second main contribution. Finally, as a third contribution, we reveal a surprising connection between the sample complexity of contrastive learning and the mistake bound in the classical notion of self-directed (online) learning [13].

## 2 Models of Contrastive Learning

Let $\mathcal{X}$ be any (finite or infinite) domain that is bounded under some metric. A concept over $\mathcal{X}$ is a subset of $\mathcal{X}$; we will identify a concept $C \subseteq \mathcal{X}$ with its indicator function on $\mathcal{X}$, given by $C(x) = 1$ if $x \in C$, and $C(x) = 0$ if $x \in \mathcal{X} \setminus C$. A concept class over $\mathcal{X}$ is a set of concepts over $\mathcal{X}$. If $C$ and $C'$ are two concepts over $\mathcal{X}$, then $\Delta(C, C')$ refers to the symmetric difference of $C$ and $C'$. Moreover, $\delta(C, C')$ is the relative size of $\Delta(C, C')$ over $\mathcal{X}$ measured under the standard uniform distribution. In case $\mathcal{X}$ is finite, this is just $|\Delta(C, C')|/|\mathcal{X}|$.

$\mathcal{S}[\mathrm{MQ}](\mathcal{C})$ denotes the minimum worst-case number of membership queries [4] for learning concepts in $\mathcal{C}$, where the worst case is taken over all concepts in $\mathcal{C}$ and the minimum is taken over all membership query learners. Moreover, $\mathrm{VCD}(\mathcal{C})$ denotes the VC dimension of $\mathcal{C}$.

Suppose a concept class $\mathcal{C}$ is fixed and an active learner asks membership queries about an unknown target concept $c \in \mathcal{C}$. In addition to the label of the queried example, the oracle might provide a contrastive example, e.g., a similar example of opposite label. We propose a more generic setting in which the active learner is aided by an oracle that provides labeled examples complementing the actively selected ones in *some* pre-defined form (using a so-called contrast set). The learner tries to identify an unknown target concept $C^* \in \mathcal{C}$ in sequential rounds. In round 1, the version space (i.e., the set of possible target concepts under consideration) is $\mathcal{C}_1 := \mathcal{C}$. Round $i$, $i \geq 1$, is as follows:

1. The learner poses a membership query consisting of an instance $x_i \in \mathcal{X}$.
2. The oracle's response is a pair $[y_i, (x_i', y_i')] \in \{0, 1\} \times (\mathcal{X} \times \{0, 1\})$ such that:
   - $y_i = C^*(x_i)$ and $y_i' = C^*(x_i')$, i.e., $y_i$ and $y_i'$ are the correct labels of $x_i$ and $x_i'$, resp., under the target concept,
   - $x_i' \in \mathrm{CS}(x_i, C^*, \mathcal{C}_i)$, where $\mathrm{CS}(x_i, C^*, \mathcal{C}_i) \subseteq \mathcal{X}$ is a set of instances called the *contrast set* of $x_i$ wrt $C^*$ and $\mathcal{C}_i$. The oracle is adversarial, in that it can pick any

$x_i' \in \mathrm{CS}(x_i, C^*, \mathcal{C}_i)$ as part of the response to query $x_i$. If $\mathrm{CS}(x, C^*, \mathcal{C}_i)$ is empty, the oracle returns a dummy response $\omega$, to convey that no contrastive example exists.

3. The version space is updated; in particular, the new version space $\mathcal{C}_{i+1}$ consists of all concepts $C$ in $\mathcal{C}_i$ for which $C(x_i) = y_i$ and:

- $\mathrm{CS}(x_i, C, \mathcal{C}_i) = \emptyset$ in case the dummy response $\omega$ is received,
- $C(x_i') = y_i'$ and $x_i' \in \mathrm{CS}(x_i, C, \mathcal{C}_i)$ otherwise.

This definition assumes that the learner knows the function mapping any tuple $(x_i, C, \mathcal{C}_i)$ to the set $\mathrm{CS}(x_i, C, \mathcal{C}_i)$. The core difference to the traditional notion of version space is that the learner can reduce the version space by knowing the contrast set mapping CS: a concept $C$, even if consistent with all seen labeled examples, is excluded from the version space if $x_i' \notin \mathrm{CS}(x_i, C, \mathcal{C}_i)$ for some $i$.

The labeled examples collected in the first $n$ rounds, together with the corresponding version spaces, then form an *interaction sequence* of length $n$ wrt CS, given by

$$[(x_1, y_1), (x_1', y_1'), \mathcal{C}_2], \ldots, [(x_n, y_n), (x_n', y_n'), \mathcal{C}_{n+1}].$$

This interaction sequence $\varepsilon$-*approximates* the target concept $C^* \in \mathcal{C}$ (for some $\varepsilon \geq 0$) iff $\mathcal{C}_{n+1} \subseteq \{C \in \mathcal{C} \mid \delta(C^*, C) \leq \varepsilon\}$. The sequence *identifies* the target concept $C^* \in \mathcal{C}$ iff $\mathcal{C}_{n+1} = \{C^*\}$.

Accordingly, we define the following notion of sample complexity.

**Definition 1** *Assume $\mathcal{X}$, $\mathcal{C}$, CS, and $\varepsilon$ are fixed. The $\varepsilon$-approximate contrast sample complexity of a learner $L$ on $C^*$ wrt $\mathcal{C}$, denoted by $\mathcal{S}_{\mathrm{CS}}(L, C^*, \mathcal{C}, \varepsilon)$, is the largest length of any valid interaction sequence $\sigma = [(x_1, y_1), (x_1', y_1'), \mathcal{C}_2], \ldots, [(x_n, y_n), (x_n', y_n'), \mathcal{C}_{n+1}]$ that $\varepsilon$-approximates $C^* \in \mathcal{C}$ and in which $L$ chooses $x_{i+1}$ on input of the history $[(x_1, y_1), (x_1', y_1')], \ldots, [(x_i, y_i), (x_i', y_i')]$, for all $i$. (Here the worst case length is taken over the sequence of possible choices of $x_i'$ made by the oracle.) Finally, $\mathcal{S}_{\mathrm{CS}}(L, \mathcal{C}, \varepsilon) = \sup_{C^* \in \mathcal{C}} \mathcal{S}_{\mathrm{CS}}(L, C^*, \mathcal{C}, \varepsilon)$, and $\mathcal{S}_{\mathrm{CS}}(\mathcal{C}, \varepsilon) = \inf_L \mathcal{S}_{\mathrm{CS}}(L, \mathcal{C}, \varepsilon)$. For exact learning, we consider only interaction sequences that identify $C^*$, and write $\mathcal{S}_{\mathrm{CS}}(\mathcal{C})$ instead of $\mathcal{S}_{\mathrm{CS}}(\mathcal{C}, \varepsilon)$.*

Our model assumes the learner has perfect knowledge of the set CS of candidates from which contrastive examples are chosen. This assumption, while often too strong in practice, is indeed realistic in program synthesis settings, where the learning algorithm knows the rules by which the oracle selects counterexamples, as discussed in Section 1 for CEGIS [2]. Here the designer has full control over both the oracle and the learner. In addition, our model is useful for further reasons:

- It can be easily modified to address additional real-world settings, e.g., one can assume that the learner's notion of CS is not identical, but similar to that used by the oracle.

- Lower bounds from our strong model (e.g., our bound in terms of the self-directed learning complexity, Theorem 17) immediately transfer to such weakened versions of the model.

A first observation is that, without limiting the choice of the contrast set CS, the contrast oracle is extremely powerful; any encoding of concepts in $\mathcal{C}$ as subsets of $\mathcal{X}$ of size at most $k$ can be used to define contrast sets witnessing a sample complexity of at most $k$:

**Proposition 2** *Let $\mathcal{C}$ be a countable concept class over a countable $\mathcal{X}$. Let $T : \mathcal{C} \to 2^{\mathcal{X}}$ be any injective function that maps every concept in $\mathcal{C}$ to a finite set of instances. Then:*

1. *There is some CS with $\mathcal{S}_{\mathrm{CS}}(\mathcal{C}) \leq \sup_{C \in \mathcal{C}} |T(C)| + 1$.*

2. *If $T(C) \nsubseteq T(C')$ for $C \neq C'$, then there is some CS such that $\mathcal{S}_{\mathrm{CS}}(\mathcal{C}) \leq \sup_{C \in \mathcal{C}} |T(C)|$.*

*In particular, if $\mathcal{X}$ is finite, then there is some CS such that $\mathcal{S}_{\mathrm{CS}}(\mathcal{C}) \leq 1 + \min\{k \mid \sum_{i=0}^{k} \binom{|\mathcal{X}|}{i} \geq |\mathcal{C}|\}$. If $\mathcal{X}$ is countably infinite, then there is some CS such that $\mathcal{S}_{\mathrm{CS}}(\mathcal{C}) = 1$.*

*Proof. (Sketch.)* Let $x_1, x_2, \ldots$ be a repetition-free enumeration of $\mathcal{X}$, and $\mathrm{CS}(x_i, C) = \{x_{j'}\}$ for $j' = \min\{j \mid j \geq i, \ x_j \in T(C)\}$. A learner sets $n_1 = 1$ and starts with iteration 1. In iteration $i$, it asks a query for $x_{n_i}$. If it receives $x_i' = x_j$ as a contrastive example, then $j \geq n_i$. The learner will then set $n_{i+1} = j + 1$ and proceed to iteration $i + 1$. See the appendix for details. $\qquad\square$

Thus, unlimited choice of the mapping CS reduces learning from a contrast oracle to decoding a smallest encoding of concepts as sample sets. Therefore, the remainder of our study focuses on more natural choices of contrast sets. In particular, when fixing a function $d : \mathcal{X} \times \mathcal{X} \to \mathbb{R}^{\geq 0}$ (interpreted as a notion of distance), two natural choices of contrast set mappings are the following:

**Minimum distance model.** Our first contrast set mapping, dubbed the *minimum distance* model, makes the oracle provide an example closest to $x$, among those that yield a different label under $C$ than $x$. For discrete $\mathcal{X}$, this translates to $\mathrm{CS}^d_{\min}(x, C) = \arg\min\{d(x, x') \mid x' \in \mathcal{X}, \; C(x') \neq C(x)\}$. For continuous $\mathcal{X}$, it can happen that no point $x' \in \mathcal{X}$ with $C(x') \neq C(x)$ attains the infimum of the values $d(x, x')$ for such $x'$.[1] To accommodate this case, we define $\mathrm{CS}^d_{\min}(x, C) = \arg\min\{d(x, x') \mid x' \in \mathcal{X}, \; x' = \lim_{i \to \infty} x'_i$ for a Cauchy sequence $(x'_i)_i$ with $C(x'_i) \neq C(x)$ for all $i\}$. (This definition may in some cases allow the oracle to return a contrastive example with the same label as that of $x$. This is helpful when no point $x' \in \mathcal{X}$ whose label differs from $C(x)$ attains the infimum of the values $d(x, x')$ for such $x'$, but mostly for complete metric spaces, where the limit of a Cauchy sequence is guaranteed to exist.[2])

**Proximity model.** For our second contrast set mapping, the learner also selects a radius $r$ independently at each round, allowing it to ask for a contrastive example within a certain vicinity of the query point $x$. For discrete $\mathcal{X}$, we then define $\mathrm{CS}^d_{\mathrm{prox}}(x, r, C) = \{x' \in \mathcal{X} \mid C(x') \neq C(x), d(x, x') \leq r\}$, i.e., the contrast set contains all $x'$ $r$-close to $x$ that have the opposite label to $x$. For continuous $\mathcal{X}$, a similar adaptation as for the minimum distance model yields $\mathrm{CS}^d_{\mathrm{prox}}(x, r, C) = \{x' \in \mathcal{X} \mid d(x, x') \leq r, \; x' = \lim_{i \to \infty} x'_i$ for a Cauchy sequence $(x'_i)_i$ with $C(x'_i) \neq C(x)$ for all $i\}$. We refer to this choice of contrast set as the *proximity* model. Note that the oracle here need not choose a contrastive example at minimum distance; *any* example in $\mathrm{CS}^d_{\mathrm{prox}}(x, r, C)$ is allowed.

Since these mappings do not depend on the version space, we dropped the argument $\mathcal{C}_i$ in $\mathrm{CS}(x, C)$.

The following useful observations are easy to prove; the proof of Remark 5 is given in the appendix.

**Remark 3** *Let $\mathcal{C}$ be a concept class over a complete space $\mathcal{X}$ with metric $d$, $x \in \mathcal{X}$, and $C \in \mathcal{C}$. Suppose there is some $x' \in \mathcal{X}$ with $C(x) \neq C(x')$ and $d(x, x') < \infty$. Then (i) $\mathrm{CS}^d_{\min}(x, C) \neq \emptyset$, and (ii) if $r \geq d(x, x')$ then $\mathrm{CS}^d_{\mathrm{prox}}(x, r, C) \neq \emptyset$.*

**Remark 4** *Suppose that $\mathrm{CS}$ and $\mathrm{CS}'$ are mappings that assign to every pair $(x, C) \in \mathcal{X} \times \mathcal{C}$ a subset of $\mathcal{X}$. Suppose, for every $(x, C) \in \mathcal{X} \times \mathcal{C}$, we have (i) $\mathrm{CS}(x, C) \subseteq \mathrm{CS}'(x, C)$, and (ii) $\mathrm{CS}(x, C) = \emptyset$ implies $\mathrm{CS}'(x, C) = \emptyset$. Then $\mathcal{S}_{\mathrm{CS}}(\mathcal{C}) \leq \mathcal{S}_{\mathrm{CS}'}(\mathcal{C})$. Moreover, for any $\varepsilon$, $\mathcal{S}_{\mathrm{CS}}(\mathcal{C}, \varepsilon) \leq \mathcal{S}_{\mathrm{CS}'}(\mathcal{C}, \varepsilon)$.*

**Remark 5** *Let $\mathcal{C}$ be a concept class over a complete space $\mathcal{X}$ with metric $d$. Then $\mathcal{S}_{\mathrm{CS}^d_{\min}}(\mathcal{C}) \leq \mathcal{S}_{\mathrm{CS}^d_{\mathrm{prox}}}(\mathcal{C}) \leq \mathcal{S}[\mathrm{MQ}](\mathcal{C})$.*

When $\mathcal{X}$ is finite, we can extend Remark 5 as follows.

**Proposition 6** *Fix $\mathcal{X}$ and $d : \mathcal{X} \times \mathcal{X} \to \mathbb{R}^{\geq 0}$. Let $S_d(x) = \{d(x, x') \mid x' \in \mathcal{X} \setminus x\}$ and $s_d = \max_{x \in \mathcal{X}} |S_d(x)|$. (Note that $s_d \leq |\mathcal{X}| - 1$ for finite $\mathcal{X}$). Then $\mathcal{S}_{\mathrm{CS}^d_{\mathrm{prox}}}(\mathcal{C}) \leq \lceil \log(s_d) \rceil \cdot \mathcal{S}_{\mathrm{CS}^d_{\min}}(\mathcal{C})$.*

*Proof.* Let $L$ learn any $C \in \mathcal{C}$ with $\leq \mathcal{S}_{\mathrm{CS}^d_{\min}}(\mathcal{C})$ queries to the minimum distance oracle w.r.t. $d$. We construct a learner $L'$ with access to the proximity oracle, using $L$ as a subroutine. Suppose $L$ poses a query $x$. Let $r_{\min} = \min_{x' \in \mathrm{CS}^d_{\min}(x, C)} d(x, x')$. First, $L'$ sorts all numbers in $S_d(x)$ in increasing order and determines the median $r$ of that list. Then $L'$ poses the query $(x, r)$. The contrastive example returned will be $\omega$ iff $d(x, x') < r_{\min}$. Hence, with a binary search using $\lceil \log(s_d) \rceil$ queries, $L'$ can determine an $x'$ with opposite label to $x$ and $d(x, x') = r_{\min}$. This $x'$ is among the admissible answers to the query by $L$. Thus, $L'$ makes $\leq \lceil \log(s_d) \rceil$ queries for every query that $L$ makes. $\square$

---

[1] For instance, if $C$ is a closed interval in $\mathbb{R}$ and the query point $x$ lies inside this interval, then there is no point outside the interval that has distance closer to $x$ than any other point outside the interval.

[2] The definition of Cauchy sequence relies on a notion of metric. In our examples in continuous domains, we will use the $\ell_1$-metric both as this metric and as the function $d$ in the superscript of $\mathrm{CS}^d_{\min}$.

| | without CS | $\mathrm{CS}^d_{\mathrm{prox}}$ | $\mathrm{CS}^d_{\mathrm{min}}$ (even for exact identification) |
|---|---|---|---|
| (a) thresholds | $\Theta(\log\frac{1}{\varepsilon})$ | $\Theta(\log\frac{1}{\varepsilon})$ | 1 |
| (b) rectangles | $\Theta(k\log(\frac{1}{\varepsilon}))$ | $\Omega(\log(\frac{1}{\varepsilon})), O(\log(\frac{k}{\varepsilon}))$ | 2 |

Table 1: Asymptotic sample complexity when $\varepsilon$-approximating (a) one-sided threshold functions, and (b) axis-aligned rectangles in $k$ dimensions, for various learning models, with the $\ell_1$-metric.

## 3 Sample Complexity Under Various Metrics

This section illustrates our notions of learning from contrast oracles under various metrics $d$. We begin with two examples on the $\ell_1$-**metric**.

**Example 1** *Consider the class of one-sided threshold functions $\mathbb{1}\{x \le \theta^*\}$ for $\theta^* \in \mathcal{X} = [0,1]$. Table 1(a) displays the asymptotic sample complexity of various models of $\varepsilon$-approximate learning. The result on learning without contrastive examples is known [25]. By a straightforward adversary argument, the sample complexity of the proximity model is in $\Omega(\log 1/\varepsilon)$. As learning without contrast oracle is no stronger than learning in the proximity model, both models here have the same asymptotic complexity. In the minimum-distance model, irrespective of the first queried instance, the contrastive example will be $\theta^\star$. Thus, the target can be identified (even exactly!) with one query.*

**Example 2** *Consider the class of all $k$-dimensional axis-aligned rectangles over $\mathcal{X} = [0,1]^k$, with the $\ell_1$-distance as metric. Table 1(b) displays the asymptotic sample complexity of various models of $\varepsilon$-approximate learning. The result on learning without contrastive examples is known, see, e.g., [14]. Let $C^\star$ be a target rectangle with the corner of lowest $\ell_1$-distance from $\mathbf{0} = (0,\ldots,0)$ being $\mathbf{x}$, and the diagonally opposed corner being $\mathbf{y}$.*
*In the minimum-distance model, let a learner first query $\mathbf{0}$. If $C^*(\mathbf{0}) = 1$ then clearly $\mathbf{x} = \mathbf{0}$. If $C^*(\mathbf{0}) = 0$ then the contrastive example $x'$ will be $\mathbf{x}$. Similarly, by querying $\mathbf{1}$, the learner can determine $\mathbf{y}$. Thus, $\mathcal{S}_{\mathrm{CS}^d_{\mathrm{min}}} \le 2$. Also, for determining $C^*$, at least two distinct positively labelled instances are required, which is impossible to acquire with one query.*
*To verify the upper bound in the proximity model, note that $d(\mathbf{x},\mathbf{0}) \le k$ for any $\mathbf{x}$. The learner starts by querying $x_1 = \mathbf{0}$ and $r_1 = \frac{k}{2}$. The contrastive example will be $\omega$ iff $r_1 < d(\mathbf{x},\mathbf{0})$. Thus, the learner determines whether $d(\mathbf{x},\mathbf{0}) \le r_1$ or not. Learning continues with a binary search until finding a radius $r_t$ such that $d(\mathbf{x},\mathbf{0}) \le r_t \le d(\mathbf{x},\mathbf{0}) + \varepsilon/2$ for $t \ge \log(2k/\varepsilon) + 1$. Let $x'_t$ be the contrastive example. We derive $d(x'_t, \mathbf{x}) \le d(x'_t, \mathbf{0}) - d(\mathbf{x},\mathbf{0}) \le r_t - d(\mathbf{x},\mathbf{0}) \le \frac{\varepsilon}{2}$ (noting that $d(x'_t, \mathbf{0}) - d(\mathbf{x},\mathbf{0}) > 0$ at step $t$). Similarly with $\log(2k/\varepsilon) + 1$ samples the learner can find a $\mathbf{z} \in [0,1]^k$ such that $d(\mathbf{z},\mathbf{y}) \le \varepsilon/2$. Therefore, the rectangle with bottom left corner $x'_t$ and top right corner $\mathbf{z}$ has error less than $\varepsilon$.*
*The lower bound on $\mathcal{S}_{\mathrm{CS}^d_{\mathrm{prox}}}(\mathcal{C}, \varepsilon)$ is inherited from that in Example 1, using the same argument.*

The next few examples concern concept classes over the Boolean domain $B_m = \{0,1\}^m$ and learning from a contrast oracle in the minimum distance model with respect to the **Hamming distance as metric**. For $I \subseteq [m]$, we denote by $\mathbf{1}_I$ the Boolean vector with 1s in positions indexed by $I$ and 0s elsewhere; let $\mathbf{1} := \mathbf{1}_{[m]}$. The notation $\mathbf{0}_I$ and $\mathbf{0}$ is understood analogously.

The next two toy examples will show that a certain metric (here the Hamming distance) may be suitable in the context of a specific concept class, but will become misleading in the minimum distance model when a slight representational change is made to the concept class. These toy examples are merely meant to illustrate the power and limitations of our abstract model in extreme situations.

**Example 3** *Let $\mathcal{X} = \{0,1\}^m$. Let $\mathcal{C}^m_{\mathrm{mmon}}$ consist of all monotone monomials, i.e., logical formulas of the form $v_{i_1} \wedge \ldots \wedge v_{i_k}$ for some pairwise distinct $i_1, \ldots, i_k$ and some $k \in \{0,\ldots,m\}$. The concept associated with such a formula contains the boolean vector $(b_1,\ldots,b_m)$ iff $b_{i_1} = \ldots = b_{i_k} = 1$ (where the empty monomial is the constant 1 function). Let $d$ be the Hamming distance. Then:*
*(1) $\mathcal{S}[\mathrm{MQ}](\mathcal{C}^m_{\mathrm{mmon}}) = m$. (2) $\mathcal{S}_{\mathrm{CS}^d_{\mathrm{min}}}(\mathcal{C}^m_{\mathrm{mmon}}) = 1$. (3) $\mathcal{S}_{\mathrm{CS}^d_{\mathrm{prox}}}(\mathcal{C}^m_{\mathrm{mmon}}) = \Theta(\log m)$.*

*Proof. (Sketch.)* (1) is folklore. For (2), a learner will first query $\mathbf{0}$, which has the label 0 unless the target is the empty monomial. The unique positively labeled example of smallest Hamming distance to the all-zeroes vector has ones exactly in $v_{i_1},\ldots,v_{i_k}$, where the target concept is $v_{i_1} \wedge \ldots \wedge v_{i_k}$.

Thus, a single query identifies the target. The upper bound in (3) follows from Proposition 6, while the lower bound is obtained via case distinction. Details are in the appendix. $\square$

Now, we will see how a small syntactic change from $\mathcal{C}^m_{\mathrm{mmon}}$ to a class $\mathcal{C}'_{\mathrm{mmon}}$ (adding an $(m+1)$st component to the boolean vectors) can make contrastive examples useless (see appendix for details).

**Example 4** *Let $\mathcal{X}' = \{0,1\}^{m+1}$. The concept class $\mathcal{C}'_{\mathrm{mmon}}$ over $\mathcal{X}'$ is defined to contain, for each $C \in \mathcal{C}^m_{\mathrm{mmon}}$, the concept $C' \in \mathcal{C}'_{\mathrm{mmon}}$ constructed via $C'(b_1,\ldots,b_m,0) = C(b_1,\ldots,b_m)$ and $C'(b_1,\ldots,b_m,1) = 1 - C(b_1,\ldots,b_m)$. No other concepts are contained in $\mathcal{C}'_{\mathrm{mmon}}$. Let $d$ be the Hamming distance. Then $\mathcal{S}[\mathrm{MQ}](\mathcal{C}'_{\mathrm{mmon}}) = \mathcal{S}_{\mathrm{CS}^d_{\min}}(\mathcal{C}'_{\mathrm{mmon}}) = \mathcal{S}_{\mathrm{CS}^d_{\mathrm{prox}}}(\mathcal{C}'_{\mathrm{mmon}}) = m$.*

While the Hamming distance is useless for the class $\mathcal{C}'_{\mathrm{mmon}}$ in Example 4, other functions $d$ might still allow for efficient learning in this setting. Such functions, however, might be rather "unnatural" as a notion of distance (arguably though, the class $\mathcal{C}'_{\mathrm{mmon}}$ in Example 4 is also somewhat "unnatural"). We will hence focus most of our study on intuitively "natural" functions $d$. When learning Boolean functions, we consider Hamming distance a natural choice.

Let $\mathcal{C}^m_{\mathrm{mon}}$ denote the class of monomials over the Boolean variables $v_1,\ldots,v_m$ (including the empty monomial which represents the constant-1 function). We assume that each Boolean variable occurs at most once (negated or not negated) in a monomial. Since monomials like $v_i \wedge \bar{v}_i$ are thus excluded, the constant-0 function does not belong to $\mathcal{C}^m_{\mathrm{mon}}$. A concept class dual to $\mathcal{C}^m_{\mathrm{mon}}$ is the class of clauses over the Boolean variables $\{v_1,\ldots,v_m\}$ (including the empty clause which represents the constant-0 function but excluding clauses like $v_i \vee \bar{v}_i$ which represent the constant-1 function). We denote this class by $\mathcal{C}^m_{\mathrm{claus}}$, and obtain the following result, the full proof of which is given in the appendix.

**Theorem 7** $\mathcal{S}_{\mathrm{CS}^d_{\min}}(\mathcal{C}^m_{\mathrm{mon}} \cup \mathcal{C}^m_{\mathrm{claus}}) = 2$, *where $d$ is the Hamming distance.*

*Proof.* (Sketch.) One first constructs a learner $L_\wedge$ for $\mathcal{C}^m_{\mathrm{mon}}$, as in the proof of Example 3, and then a learner $L_\vee$ for $\mathcal{C}^m_{\mathrm{claus}}$ by dualization from $L_\wedge$. It suffices to describe a learner $L$ that maintains simulations of $L_\wedge$ and $L_\vee$ and *either* comes in both simulations to the same conclusion about the target concept *or* realizes an inconsistency in one of the simulations, which can then be aborted. $\square$

For $\mathcal{S}_{\mathrm{CS}^d_{\mathrm{prox}}}$, an upper bound is obtained from Proposition 6 and Theorem 7, noting that the Hamming distance can take only $m$ distinct values for pairs $(x,x')$ with $x \neq x'$.

**Corollary 8** $\mathcal{S}_{\mathrm{CS}^d_{\mathrm{prox}}}(\mathcal{C}^m_{\mathrm{mon}}) \cup \mathcal{S}_{\mathrm{CS}^d_{\mathrm{prox}}}(\mathcal{C}^m_{\mathrm{claus}}) \leq 2\lceil \log m \rceil$, *where $d$ is the Hamming distance.*

We conclude this subsection with an example of a somewhat natural concept class for which the minimum distance model under the Hamming distance is not very powerful.

**Example 5** *Let $\mathcal{X} = \{0,1\}^m$ and let $\mathcal{C}_{\mathrm{par}}$ be the class of parity functions, i.e., logical formulas of the form $v_{i_1} \oplus \ldots \oplus v_{i_k}$ for some pairwise distinct $i_1,\ldots,i_k$ and some $k \in \{0,\ldots,m\}$. Then $\mathcal{S}[\mathrm{MQ}](\mathcal{C}_{\mathrm{par}}) = m$ and $m - 1 \leq \mathcal{S}_{\mathrm{CS}^d_{\min}}(\mathcal{C}_{\mathrm{par}}) = \mathcal{S}_{\mathrm{CS}^d_{\mathrm{prox}}}(\mathcal{C}_{\mathrm{par}}) \leq m$.*

*Proof.* $\mathcal{S}[\mathrm{MQ}](\mathcal{C}_{\mathrm{par}}) = m$ is trivial, and immediately yields $\mathcal{S}_{\mathrm{CS}^d_{\min}}(\mathcal{C}_{\mathrm{par}}) \leq m$, $\mathcal{S}_{\mathrm{CS}^d_{\mathrm{prox}}}(\mathcal{C}_{\mathrm{par}}) \leq m$. Suppose the target concept is $c^* = v_{i_1} \oplus \ldots \oplus v_{i_k}$. For any instance $\mathbf{x}$, we have $d(\mathbf{x},\mathbf{x}') = 1$ and $c^*(\mathbf{x}') \neq c^*(\mathbf{x}')$, where $\mathbf{x}'$ results from $\mathbf{x}$ by flipping the value of component $i_1$. When always using a query radius $r_t = 1$, the proximity model is thus equivalent to the minimum distance model. The lower bound is witnessed by an oracle that contrasts any $\mathbf{x}_t$ with the example $\mathbf{x}'_t$ resulting from $\mathbf{x}_t$ by flipping the component $i_1$. This provides no information about variables other than $v_{i_1}$. $\square$

We conclude this section with the **discrete metric** $d_0$, defined by $d_0(x,x') = 0$ if $x = x'$, and $d_0(x,x') = 1$ if $x \neq x'$. Now $\mathrm{CS}^d_{\min}(x,C) \subseteq \mathrm{CS}^{d_0}_{\min}(x,C)$ for every $(x,C) \in \mathcal{X} \times \mathcal{C}$ and every $d : \mathcal{X} \times \mathcal{X} \to \mathbb{R}^{\geq 0}$. Moreover, $\mathrm{CS}^d_{\min}(x,C) = \emptyset$ implies $\mathrm{CS}^{d_0}_{\min}(x,C) = \emptyset$. Remark 4 thus yields:

**Corollary 9** *For every concept class $\mathcal{C}$, the term $\mathcal{S}_{\mathrm{CS}^d_{\min}}(\mathcal{C})$ is maximized, if the underlying function $d : \mathcal{X} \times \mathcal{X} \to \mathbb{R}^{\geq 0}$ equals the discrete metric $d_0$.*

In what follows, if $O$ is a list of oracles, we use $\mathcal{S}[O]$ to denote the sample complexity of learning with access to all oracles in $O$. Let $\mathrm{EX}^+$ be an oracle returning a positive example $x$, i.e., an $x \in \mathcal{X}$

with $C^*(x) = 1$ (or a dummy response if no such $x$ exists). When called several times, the oracle may repeatedly present the same example. Let $\text{EX}^-$ be the analogous oracle for a negative example. The following result says that, under the metric $d_0$, the contrast oracle does not provide significantly more information than the membership oracle (although, for some special concept classes, this small extra information can make a big difference). The proof is given in the appendix.

**Theorem 10** $\mathcal{S}[\text{EX}^+, \text{EX}^-, \text{MQ}](\mathcal{C}) - 2 \leq \mathcal{S}_{\text{CS}_{\min}^{d_0}}(\mathcal{C}) \leq \mathcal{S}[\text{EX}^+, \text{EX}^-, \text{MQ}](\mathcal{C})$.

**Example 6** *Consider $\mathcal{C}$ to be the class of all singleton concepts over a finite domain $\mathcal{X}$ where $n := |\mathcal{X}| \geq 2$. It is known that $\mathcal{S}[\text{MQ}](\mathcal{C}) = n - 1$. But a single call of the oracle $\text{EX}^+$ reveals the identity of the target concept. It follows that $\mathcal{S}[\text{EX}^+](\mathcal{C}) = \mathcal{S}[\text{EX}^+, \text{EX}^-, \text{MQ}](\mathcal{C}) = \mathcal{S}_{\text{CS}_{\min}^{d_0}}(\mathcal{C}) = 1$.*

# 4 Lower Bounds

In this section, we will derive some lower bounds on the sample complexity in our models of learning from contrastive examples. All our results here are for finite instance spaces.

## 4.1 A Metric-Independent Lower Bound

Self-directed learning [13] is a model of online learning, in which the sequence of data to be labeled is chosen by the learner as opposed to the environment. The learner gets immediate feedback on every single prediction, which allows it to adapt its data selection to the version space sequentially. It is then assessed in terms of the number of incorrect predictions it makes across all data instances in $\mathcal{X}$; as usual, we consider the worst case over all concepts in the underlying class $\mathcal{C}$.

**Definition 11 ([13])** *Let $\mathcal{C}$ be a concept class over a finite domain $\mathcal{X}$, $n = |\mathcal{X}|$. A self-directed learner $L$ for $\mathcal{C}$ interacts with an oracle in $n$ rounds. In each round, $L$ selects a not previously selected instance $x \in \mathcal{X}$ and predicts a label $b$ for $x$. The oracle knows the target $C^* \in \mathcal{C}$ and returns the label $C^*(x)$; if $b \neq C^*(x)$, the learner incurs a mistake. $L$ proceeds to the next round until all instances in $\mathcal{X}$ have been labeled. The cost of $L$ on $C^*$ is the number of mistakes made in these $n$ rounds. The cost of $L$ on $\mathcal{C}$ is the maximum cost of $L$ on any $C \in \mathcal{C}$.*
*The self-directed learning complexity of $\mathcal{C}$, denoted by $\text{SD}(\mathcal{C})$, is the smallest number $c$ such that there exists a self-directed learner for $\mathcal{C}$ whose cost on $\mathcal{C}$ is $c$.*

This complexity notion was analyzed relative to other learning-theoretic parameters [6, 11], and extended to infinite concept classes and multi-class settings [10]. Interestingly, SD can be used to derive a metric-independent lower bound on the sample complexity of the minimum-distance model.

**Theorem 12** *If $\mathcal{C}$ is defined over a finite $\mathcal{X}$, and $d : \mathcal{X} \times \mathcal{X} \to \mathbb{R}^{\geq 0}$, then $\mathcal{S}_{\text{CS}_{\min}^d}(\mathcal{C}) \geq \lceil \text{SD}(\mathcal{C})/2 \rceil$.*

*Proof.* Fix $d : \mathcal{X} \times \mathcal{X} \to \mathbb{R}^{\geq 0}$, and let $L$ be a learner with access to a minimum-distance oracle w.r.t. $d$, which makes at most $\mathcal{S}_{\text{CS}_{\min}^d}(\mathcal{C})$ queries to identify any $C \in \mathcal{C}$. We construct a self-directed learner $L'$ for $\mathcal{C}$ which calls $L$ as a subroutine. If $L$ poses a query for $x$, then $L'$ predicts 0 on $x$. After receiving feedback, $L'$ passes on the correct label $b$ for $x$ to $L$. In addition, $L'$ must provide a contrastive example for $x$, which is obtained as follows: $L'$ sorts all instances in $\mathcal{X}$ in increasing order w.r.t. their distance from $x$ as measured by $d$. It then predicts the label $b$ for instances in this ordered list until it observes the first mistake, say for $x'$. Then $x'$ must be a closest instance to $x$ with a label opposite from $b$. Hence, $x'$ is passed on to $L$ as a contrastive example. If no such $x'$ is found, then $L'$ gives the response $\omega$ to $L$. This proceeds until all instances in $\mathcal{X}$ have been labeled.

Each time $L'$ calls $L$, it incurs at most two mistakes, namely one for its prediction on the instance $x$ queried by $L$, and one for its prediction on the contrastive instance $x'$. In total, thus $L'$ makes at most $2t$ mistakes, where $t$ is the number of queries posed by $L$. We obtain the desired bound. $\qquad\square$

**Remark 13** *For each $m \in \mathbb{N}$, there is a concept class $\mathcal{C}_m$ with $\text{VCD}(\mathcal{C}_m) = 2$ and $\text{SD}(\mathcal{C}_m) \geq m$ [18]. In particular, there is no relationship between $\text{VCD}$ and $\text{SD}$. Thus, no matter which function $d : \mathcal{X} \times \mathcal{X} \to \mathbb{R}^{\geq 0}$ is chosen, there exists no upper bound on $\mathcal{S}_{\text{CS}_{min}^d}$ as a function of $\text{VCD}$.*

We will see multiple cases in which the lower bound provided by Theorem 12 is not met. As a small warm-up example, there are finite classes of VC dimension 1 for which $\mathcal{S}_{\mathrm{CS}^d_{\min}}(\mathcal{C}) = 2$, while every class of VC dimension 1 satisfies $\mathrm{SD} = 1$ (see the appendix for proof details):

**Example 7** *Let $\mathcal{X} = \{x_1, x_2, x_3\}$. Let $\mathcal{C}$ consist of the concepts $\{x_1, x_3\}$, $\{x_3\}$, $\{x_2, x_3\}$, and $\{x_2\}$. Then $\mathcal{C}$ satisfies $\mathrm{SD}(\mathcal{C}) = \mathrm{VCD}(\mathcal{C}) = 1$, but $\mathcal{S}_{\mathrm{CS}^d_{\min}}(\mathcal{C}) \geq 2$ for any $d : \mathcal{X} \times \mathcal{X} \to \mathbb{R}^{\geq 0}$.*

However, for classes of VC dimension 1, we cannot increase the gap between SD and $\min_d \mathcal{S}_{\mathrm{CS}^d_{\min}}$:

**Theorem 14** *For any finite $\mathcal{C}$ with $\mathrm{VCD}(\mathcal{C}) = 1$ there exists a metric $d$ with $\mathcal{S}_{\mathrm{CS}^d_{\min}}(\mathcal{C}) \leq 2$.*

The proof (see appendix) uses the fact that every $x \in \mathcal{X}$ has at most one label $b \in \{0, 1\}$ such that $(x, b)$ "teaches" a unique $C \in \mathcal{C}$ in the model of recursive teaching [11]. The instances in $\mathcal{X}$ can be partitioned according to these labels. This partitioning is used to define a metric that places instances in the same part closer to one another than instances in different parts. Now a learner choosing one specific query from each of the two parts will receive enough information to identify the target.

## 4.2 Lower Bounds Using the Hamming Distance

Let $\mathrm{DL}_m$ be the class of 1-decision lists over the Boolean variables $v_1, \ldots, v_m$. Each list is of shape

$$\mathcal{L} = [(\ell_1, b_1), \ldots, (\ell_z, b_z), b_{z+1}] \ , \tag{1}$$

where $z \geq 0$, $\ell_i \in \{v_1, \bar{v}_1, \ldots, v_m, \bar{v}_m\}$ and $b_i \in \{0, 1\}$. We assume that each variable occurs at most once in $\mathcal{L}$ (negated or not negated). We say that a point $\mathbf{a} \in B_m$ is *absorbed by item $k$ of $\mathcal{L}$* if the literal $\ell_k$ applied to $\mathbf{a}$ evaluates to 1 while the literals $\ell_1, \ldots, \ell_{k-1}$ applied to $\mathbf{a}$ evaluate to 0. The list $\mathcal{L}$ represents the following Boolean function $f_{\mathcal{L}}$: (i) If $\mathbf{a}$ is absorbed by the $k$'th item of $\mathcal{L}$, then $f_{\mathcal{L}}(\mathbf{a}) = b_k$. (ii) If $\mathbf{a}$ is absorbed by none of the $z$ items of $\mathcal{L}$, then $f_{\mathcal{L}}(\mathbf{a}) = b_{z+1}$.

The following lemma observes a structural property of a concept class $\mathcal{C}$ which allows us to lower-bound the sample complexity in the minimum-distance model (with the Hamming distance) using the membership query complexity of a certain subclass of $\mathcal{C}$.

**Lemma 15** *Suppose that $\mathcal{C}$ is a concept class over $B_m$, and $\mathcal{C}'$ is a subclass of $\mathcal{C}$ with the properties*
*(P1) For each $\mathbf{a} \in B_m$ with $a_m = y_m$ and each $C \in \mathcal{C}'$, we have that $C(\mathbf{a}) = 1$.*
*(P2) For each $\mathbf{a}$ with $a_m = y'_m$ and $a_{m-1} = y_{m-1}$ and each $C \in \mathcal{C}'$, we have that $C(\mathbf{a}) = 0$.*
*For $i \in \{m, m-1\}$, here $y_i \in \{0, 1\}$ is fixed, $y'_i = 1 - y_i$, and $\mathcal{C}''$ is the set of all concepts of shape*

$$(a_1, \ldots, a_{m-2}) \mapsto C(a_1, \ldots, a_{m-2}, y'_{m-1}, y'_m) \text{ for } C \in \mathcal{C}' \ .$$

*Then $\mathcal{S}_{\mathrm{CS}^d_{\min}}(\mathcal{C}) \geq \mathcal{S}_{\mathrm{CS}^d_{\min}}(\mathcal{C}') \geq \mathcal{S}[\mathrm{MQ}](\mathcal{C}'')$, where $d$ is the Hamming distance.*

*Proof. (Sketch.)* Suppose $L$ learns $\mathcal{C}'$ from $q$ queries to a contrast oracle in the minimum distance model. We show that $L$ can be transformed into $L'$ which learns $\mathcal{C}''$ from at most $q$ queries to a membership oracle. To this end, let $C''$ be the target concept in $\mathcal{C}''$ and let $C'$ with $C'(\mathbf{a}y'_{m-1}y'_m) = C''(\mathbf{a})$ be the corresponding concept in $\mathcal{C}'$. Note that $L$ has not uniquely identified $C'$ in $\mathcal{C}'$ as long as the subfunction $C''$ is not uniquely identified in $\mathcal{C}''$. $L'$ can therefore identify $C''$ in $\mathcal{C}''$ by maintaining a simulation of $L$ until $L$ has identified $C'$ in $\mathcal{C}$. Details are given in the appendix. $\square$

This lemma can be used, for example, to prove $\mathcal{S}_{\mathrm{CS}^d_{\min}}(\mathrm{DL}_m) \geq 2^{m-2} - 1$, where $d$ is the Hamming distance. Our proof of this statement, however, establishes an even stronger result. To formulate this stronger result, we first need to introduce some notation.

Consider a decision list $\mathcal{L}$ of the form (1). We say that $\mathcal{L}$ has $k$ *label alternations* if the number of distinct indices $i \in [z]$ such that $b_{i+1} \neq b_i$ equals $k$. Let $\mathrm{DL}_m^k$ be the subclass of $\mathrm{DL}_m$ which contains all decision lists with at most $k$ label alternations. It is well known that (i) $\mathrm{DL}_m^0$ contains only the constant-1 and the constant-0 function, and (ii) $\mathrm{DL}_m^1 = \mathcal{C}_{\mathrm{mon}}^m \cup \mathcal{C}_{\mathrm{claus}}^m$. Now the proof of the following claim builds on Lemma 15; see the appendix for details.

**Corollary 16** *$\mathcal{S}_{\mathrm{CS}^d_{\min}}(\mathrm{DL}_m^2) \geq 2^{m-2} - 1$, where $d$ is the Hamming distance.*

Interestingly, this bound on $\mathcal{S}_{\mathrm{CS}^d_{\min}}(\mathrm{DL}_m)$ (using Hamming distance) is not an effect of our metric-independent lower bound in terms of SD (Thm. 12). As seen below, $\mathcal{S}_{\mathrm{CS}^d_{\min}}(\mathrm{DL}_m)$ asymptotically exceeds $\mathrm{SD}(\mathrm{DL}_m)$. Thus, the lower bound from Thm. 12 is not always asymptotically tight.

For $\mathrm{DL}_m$, we saw that the sample complexity in the minimum Hamming distance model is exponential in $m$. By contrast, SD is at most quadratic in $m$. To prove this, let a *block* in $[(\ell_1, b_1), \ldots, (\ell_z, b_z), b_{z+1}]$ be any maximal substring $(\ell_i, b_i), \ldots, (\ell_{i+j}, b_{i+j})$ with $b_i = \ldots = b_{i+j}$.

**Theorem 17** *There exists a self-directed learner for $\mathrm{DL}_m$ that makes, on any target list $C^* \in \mathrm{DL}_m$, at most $4km$ mistakes where $k$ is the number of blocks in $C^*$.*

*Proof. (Sketch.)* For each of the $2m$ literals, 2 queries suffice to determine whether it occurs in the first block of the target list. Iteratively, one consumes $\leq 4m$ queries per block; see the appendix. $\square$

We now present a second example of a natural class of Boolean functions for which SD is asymptotically smaller than $\mathcal{S}_{\mathrm{CS}^d_{\min}}(\mathcal{C})$, where $d$ is the Hamming distance.

**Definition 18** *Fix $s, z, m \in \mathbb{N}$. An $s$-term $z$-MDNF of size $m$ is a function $f : \{0,1\}^m \to \{0,1\}$ of the form $f(v_1, v_2, \cdots, v_m) = M_1 \vee M_2 \vee \cdots \vee M_s$, where each $M_i$ is a monotone monomial with at most $z$ literals. We use $\mathcal{C}^{m,s,z}_{\mathrm{MDNF}}$ to refer to the class of all $s$-term $z$-MDNF of size $m$.*

**Theorem 19 (largely due to [13, 1])** *Let $d$ be the Hamming distance. Then $\mathrm{SD}(\mathcal{C}^{m,s,z}_{\mathrm{MDNF}}) = s$, while $\mathcal{S}_{\mathrm{CS}^d_{\min}} \geq \mathcal{S}[\mathrm{MQ}](\mathcal{C}^{m-2,s-1,z-1}_{\mathrm{MDNF}})$, which is at least*

$$(z-1)(s-1)\log(m-2) + \alpha\,,$$

*where $\alpha = \left(\frac{z-1}{s-1}\right)^{s-2}$ if $z > s$, and $\alpha = \left(\frac{2s-2}{z-1}\right)^{(z-1)/2}$ if $z \leq s$.*

*Proof.* See [13] for $\mathrm{SD}(\mathcal{C}^{m,s,z}_{\mathrm{MDNF}}) = s$. [1] showed the lower bound on $\mathcal{S}[\mathrm{MQ}](\mathcal{C}^{m-2,s-1,z-1}_{\mathrm{MDNF}})$.

Let $\mathcal{C}'$ be the subclass of $\mathcal{C}^{m,s,z}_{\mathrm{MDNF}}$ with concepts of the form $M_1 \vee \ldots \vee M_{s-1} \vee v_m$ where $v_{m-1} \in M_i$ for $1 \leq i \leq s-1$. Then $\mathcal{C}'$ satisfies properties (P1) and (P2) of Lemma 15 with $y_m = 1$, and $y_{m-1} = 0$. Then $\mathcal{C}'' = \mathrm{MDNF}_{m-2,z-1,s-1}$, and the result follows from Lemma 15. $\square$

Theorem 19 reveals a gap between SD and the sample complexity of the minimum distance model under the Hamming distance. While we do not know of any "natural" function $d : \mathcal{X} \times \mathcal{X} \to \mathbb{R}^{\geq 0}$ with which to close this gap, it can be closed if we allow the learner and the contrast oracle to share a new function $d$ after every query. By $\mathcal{S}_{\mathrm{CS}^{(d^t)}_{\min}}$, we denote the sample complexity resulting from interaction sequences at which, in step $t$, the set CS is defined using the distance function $d^t$. The same proof as for Theorem 12 shows that our lower bound in terms of SD (and thus the lack of an upper bound in terms of VCD, cf. Remark 13) persists despite this strengthening of the model.

**Theorem 20** *Fix $(d_t)_{t=1}^{\infty}$ and $\mathcal{C}$ over a finite domain $\mathcal{X}$. Then $\mathcal{S}_{\mathrm{CS}^{(d^t)}_{\min}}(\mathcal{C}) \geq \lceil \mathrm{SD}(\mathcal{C})/2 \rceil$.*

However, the dynamic distance function approach helps to overcome obstacles noted in Theorem 19. One natural way to define dynamically changing functions $d^t$ is to make them dependent on the version space. Defining $d_{\mathcal{C}'}(x, x') := |\{C \in \mathcal{C}' : C(x) \neq C(x')\}|/|\mathcal{C}'|$ for any $\mathcal{C}' \subset \mathcal{C}$, and recalling that $\mathcal{C}_t$ refers to the version space obtained from $\mathcal{C}$ after $t$ queries, we get (see appendix for a proof):

**Theorem 21** $\mathcal{S}_{CS^{(d_{\mathcal{C}_t})}_{\min}}(\mathcal{C}^{m,s,z}_{\mathrm{MDNF}}) \leq s = \mathrm{SD}(\mathcal{C}^{m,s,z}_{\mathrm{MDNF}})$, *where $\mathcal{C}_t = (\mathcal{C}^{m,s,z}_{\mathrm{MDNF}})_t$ for any $t$.*

## 5  Discussion and Conclusions

We proposed and analyzed a generic framework for active learning with contrastive examples. In studying the sample complexity in two of its instantiations (the minimum distance and the proximity model), we observed interesting connections to other models of query learning, and most notably to self-directed learning, which also led to a new result on self-directed learning of decision lists (Theorem 17). Our definition allows modelling contrastive learning with a multitude of contrast set rules, and it can easily be extended to encompass passive learners receiving contrastive examples.

Our framework allows the learner to reason with perfect knowledge about the choice of the contrast set. While this is unrealistic in some practical situations, it has applications in formal methods, where the contrast oracle is a computer program that could be simulated by the learner in order to refute a hypothesis (a concept $C$ can be excluded if the oracle does not provide the same counterexamples when $C$ is the target). Moreover, the learner's knowledge of the mapping CS implicitly connects our framework to *self-supervised learning*. Consider, for example, passive learning under the minimum distance model. Here the learner is passively sampling examples $(x, y)$, which are supplemented with contrastive examples from $\mathrm{CS}_{\min}^d$. Following our model, the learner knows that the contrastive example $x'$ to $x$ is the closest one to $x$ that has a label different from $y$. In practice, this can be implemented as a self-supervised mechanism in which the learner internally labels with $y$ all data points that are strictly closer to $x$ than $x'$.

In future work, the condition that the learner has perfect knowledge of the set CS can be softened so as to encompass more learning settings. To this end, note that contrastive examples can still be very helpful in case of a mismatch between the learner's assumption on the oracle and the actual strategy of the oracle. Consider for instance the case of learning halfspaces: (a) First assume the oracle is the perfect minimum-distance oracle, with respect to the $\ell_2$ metric. A learner with perfect knowledge of the oracle will succeed after just a single query $x_1$, if it conjectures the halfspace perpendicular to the vector $x_1' - x_1$ at the point $x_1'$, and consistent with the observed labels for $x_1$ and $x_1'$. This is because the decision boundary of the target concept is perpendicular to the vector $x_1' - x_1$ at the point $x_1'$. Note that this holds even if the learner picks the first query $x_1$ at random. (b) Now, suppose the same learner still expects the minimum-distance oracle, but the oracle returns a point $x_1''$ whose distance from the true minimum-distance counterexample may be up to $\gamma$. In this case, it is not hard to show that the classification error of the halfspace perpendicular to $x_1'' - x_1$ at $x_1''$ is at most

$$\frac{\cos^{-1}\left(\frac{r}{r+\gamma}\right)}{\pi}.$$

Here $r$ refers to the distance from $x_1$ to the target hyperplane. Thus, even when the initial point $x_1$ is chosen at random, the expected classification error remains linear in $\gamma$, with just a single contrastive example.

There are also scenarios where receiving an instance with the exact minimum distance is not necessarily optimal. For example, when learning a halfspace in the presence of label noise, the perfect minimum-distance oracle would provide the learner with outliers that may be far away from the decision boundary. Here a softer variant of the minimum-distance model may be preferable. One such alternative is an oracle that returns the closest instance $x'$ to the queried instance $x$, such that the majority of the $k$ nearest neighbors of $x'$ (for some parameter $k$) have a different label than $x$.

## Acknowledgments and Disclosure of Funding

The authors would like to thank Valentio Iverson and Mohamadsadegh Khosravani for insightful discussions. Moreover, thanks are due to the anonymous reviewers for their helpful feedback on an earlier version of this paper.

Y. Chen acknowledges the support of the National Science Foundation under Grant no. IIS 2313131. S. Zilles was supported through a Canada CIFAR AI Chair at the Alberta Machine Intelligence Institute (Amii), through a Natural Sciences and Engineering Research Council (NSERC) Canada Research Chair, through the New Frontiers in Research Fund (NFRF) under grant no. NFRFE-2023-00109 and through the NSERC Discovery Grants program under grant no. RGPIN-2017-05336.

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

# A Appendix

This appendix provides the proofs omitted from the main paper.

**Proposition 2** *Let $\mathcal{C}$ be a countable concept class over a countable $\mathcal{X}$. Let $T : \mathcal{C} \to 2^{\mathcal{X}}$ be any injective function that maps every concept in $\mathcal{C}$ to a finite set of instances. Then:*

1. *There is some CS with $\mathcal{S}_{\mathrm{CS}}(\mathcal{C}) \leq \sup_{C \in \mathcal{C}} |T(C)| + 1$.*

2. *If $T(C) \not\subseteq T(C')$ for $C \neq C'$, then there is some CS such that $\mathcal{S}_{\mathrm{CS}}(\mathcal{C}) \leq \sup_{C \in \mathcal{C}} |T(C)|$.*

*In particular, if $\mathcal{X}$ is finite, then there is some CS such that $\mathcal{S}_{\mathrm{CS}}(\mathcal{C}) \leq 1 + \min\{k \mid \sum_{i=0}^{k} \binom{|\mathcal{X}|}{i} \geq |\mathcal{C}|\}$. If $\mathcal{X}$ is countably infinite, then there is some CS such that $\mathcal{S}_{\mathrm{CS}}(\mathcal{C}) = 1$.*

*Proof.* Let $T : \mathcal{C} \to \mathcal{X}$ be such that $T(C) \neq T(C')$ whenever $C \neq C'$. Let $x_1, x_2, \ldots$ be a fixed repetition-free enumeration of all the elements in $\mathcal{X}$.

To prove statement 1, define CS by $\mathrm{CS}(x_i, C) = \{x_{j'}\}$ for $j' = \min\{j \mid j \geq i, \ x_j \in T(C)\}$. Now an active learner operates as follows. Initially, it sets $n_1 = 1$ and starts with iteration 1. In iteration $i$, it asks a query for $x_{n_i}$. If it receives $x'_i = x_j$ as a contrastive example, then note that $j \geq n_i$. The learner will then set $n_{i+1} = j + 1$ and proceed to iteration $i + 1$.

Upon its $i$th query, the learner will find out the label for $x_{n_i}$ (which is irrelevant) and the $i$th element of $T(C)$ in the enumeration $x_1, x_2, \ldots$. After $|T(C)|$ queries, the learner has seen all elements from $T(C)$, and will receive a dummy response for its next query. From its responses, it has now inferred that its target is a concept that is mapped to $T(C)$ by mapping $T$. By injectivity of $T$, this means that the learner has uniquely identified $C$. This proves statement 1. Since an injective mapping $T$ using all subsets of $\mathcal{X}$ of size at most $k^*$ can encode $\sum_{i=0}^{k^*} \binom{|\mathcal{X}|}{i}$ concepts, such $T$ would yield $\mathcal{S}_{\mathrm{CS}}(\mathcal{C}) \leq 1 + \min\{k \mid \sum_{i=0}^{k} \binom{|\mathcal{X}|}{i} \geq |\mathcal{C}|\}$.

For statement 2, note that, if $T(C) \not\subseteq T(C')$ for distinct $C, C' \in \mathcal{C}$, then the learner can infer $C$ correctly after having seen all elements in $T(C)$. With this result, for countably infinite $\mathcal{X}$, any mapping $T$ identifying each concept with a different instance in $\mathcal{X}$, results in $\mathcal{S}_{\mathrm{CS}}(\mathcal{C}) = 1$. $\qquad\square$

**Remark 5** *Let $\mathcal{C}$ be a concept class over a complete space $\mathcal{X}$ with metric $d$. Then $\mathcal{S}_{\mathrm{CS}_{\min}^d}(\mathcal{C}) \leq \mathcal{S}_{\mathrm{CS}_{\mathrm{prox}}^d}(\mathcal{C}) \leq \mathcal{S}[\mathrm{MQ}](\mathcal{C})$.*

*Proof.* The second inequality holds by definition. For the first inequality, consider any $r > 0$ and $(x, C) \in \mathcal{X} \times \mathcal{C}$. Due to Remark 3, $\mathrm{CS}_{\min}^d(x, C) = \emptyset$ iff there is no $x'$ with $C(x') \neq C(x)$. Moreover, $\mathrm{CS}_{\mathrm{prox}}^d(x, r, C) = \emptyset$ iff there are no points with different label converging to a point at distance at most $r$ from $x$. Therefore, either $\mathrm{CS}_{\mathrm{prox}}^d(x, r, C) = \emptyset$, or $\mathrm{CS}_{\mathrm{prox}}^d(x, r, C) \supseteq \mathrm{CS}_{\min}^d(x, C) \neq \emptyset$. In the former case, the contrastive oracle in the minimum distance model also conveys that there are no points with different label converging to a point at distance at most $r$ from $x$. Consequently, the contrastive oracle in the proximity model does not provide extra information. In the later case, any contrastive example given by the contrastive oracle in the minimum distance model can be also given by the contrastive oracle in the proximity model. Again, the contrastive oracle in the proximity model does not provide extra information. $\qquad\square$

**Example 3** *Let $\mathcal{X} = \{0,1\}^m$. Let $\mathcal{C}_{\mathrm{mmon}}^m$ consist of all monotone monomials, i.e., logical formulas of the form $v_{i_1} \wedge \ldots \wedge v_{i_k}$ for some pairwise distinct $i_1, \ldots, i_k$ and some $k \in \{0, \ldots, m\}$. The concept associated with such a formula contains the boolean vector $(b_1, \ldots, b_m)$ iff $b_{i_1} = \ldots = b_{i_k} = 1$ (where the empty monomial is the constant 1 function). Let $d$ be the Hamming distance. Then: (1) $\mathcal{S}[\mathrm{MQ}](\mathcal{C}_{\mathrm{mmon}}^m) = m$. (2) $\mathcal{S}_{\mathrm{CS}_{\min}^d}(\mathcal{C}_{\mathrm{mmon}}^m) = 1$. (3) $\mathcal{S}_{\mathrm{CS}_{\mathrm{prox}}^d}(\mathcal{C}_{\mathrm{mmon}}^m) = \Theta(\log m)$.*

*Proof.* It remains to prove the lower bound from Claim (3).

Let $(\mathbf{x}, r)$ be the first query point of the learner. Let $w$ be the Hamming weight of $\mathbf{x}$. The logarithmic lower bound is now obtained by an adversary argument.

If $w > m/3$, then the oracle returns $(\mathbf{x}, 1)$ and $(\mathbf{x}', 0)$ where $\mathbf{x}'$ is obtained from $\mathbf{x}$ by flipping one of the 1-entries in $\mathbf{x}$ to 0. Note that, if $j_1, \ldots, j_{w-1}$ denote the positions of the 1-entries in $\mathbf{x}'$, then the index set associated with the variables in $C^*$ could be any subset of $\{j_1, \ldots, j_{w-1}\}$.

If $r \geq m/3$ and $w \leq m/3$, then we may assume without loss of generality that $r \leq m - w$. The oracle returns $(\mathbf{x}, 0)$ and $(\mathbf{x}', 1)$ where $\mathbf{x}'$ is obtained by adding $r$ ones to $\mathbf{x}$, Note that, if $j_1, \ldots, j_r$ denote the positions of the additional ones, then the index set associated with the variables in $C^*$ could be any subset of $\{j_1, \ldots, j_r\}$.

If $r < m/3$ and $w \leq m/3$, then the oracle returns $(\mathbf{x}, 0)$ and $\omega$. Let $J$ with $|J| = w$ be the set of indices $j$ such that $x_j = 1$. Fix a set $K \subseteq [m]$ such that $|K| = r + 1$ and $K \cap J = \emptyset$. Let $I = [m] \setminus (J \cup K)$ be the set of the remaining indices in $[m]$. Let us make the commitment that the variables $x_k$ with $k \in K$ are contained in $C^*$. Even if the learner knew about this commitment, it would still have to determine the $I$-indices of the remaining variables in $C^*$. These indices could form an arbitrary subset of $I$.

In any case the learner can reduce the search space for the index set of $C^*$ not more than by a factor of 3. Hence the learner requires $\Omega(\log m)$ queries for the exact identification of $C^*$. $\qquad \square$

**Example 4** *Let $\mathcal{X}' = \{0,1\}^{m+1}$. The concept class $\mathcal{C}'_{\text{mmon}}$ over $\mathcal{X}'$ is defined to contain, for each $C \in \mathcal{C}^m_{\text{mmon}}$, the concept $C' \in \mathcal{C}'_{\text{mmon}}$ constructed via $C'(b_1, \ldots, b_m, 0) = C(b_1, \ldots, b_m)$ and $C'(b_1, \ldots, b_m, 1) = 1 - C(b_1, \ldots, b_m)$. No other concepts are contained in $\mathcal{C}'_{\text{mmon}}$. Let $d$ be the Hamming distance. Then $\mathcal{S}[\mathrm{MQ}](\mathcal{C}'_{\text{mmon}}) = \mathcal{S}_{\mathrm{CS}^d_{\min}}(\mathcal{C}'_{\text{mmon}}) = \mathcal{S}_{\mathrm{CS}^d_{\text{prox}}}(\mathcal{C}'_{\text{mmon}}) = m$.*

*Proof.* A membership query learner using $m$ queries will work as described in the proof of Example 3, yet will leave the $(m+1)$st component of every queried vector equal to 0. Thus, the learner will identify which vectors of the form $(b_1, \ldots, b_m, 0)$ belong to the target concept. The latter immediately implies which vectors of the form $(b_1, \ldots, b_m, 1)$ belong to the target concept. Again, by a standard information-theoretic argument, fewer membership queries will not suffice in the worst case. Thus $\mathcal{S}[\mathrm{MQ}](\mathcal{C}'_{\text{mmon}}) = m$.

Upon asking any query $(b_1, \ldots, b_m, b_{m+1})$, a learner in the minimum distance model will receive the correct answer, plus the contrastive example $(b_1, \ldots, b_m, 1 - b_{m+1})$. This contrastive example does not provide any additional information to the learner. Hence, the learner has to ask as many queries as an MQ-learner, i.e., $\mathcal{S}_{\mathrm{CS}^d_{\min}}(\mathcal{C}'_{\text{mmon}}) = m$.

Finally, according to Remark 5, $\mathcal{S}_{\mathrm{CS}^d_{\min}} \leq \mathcal{S}_{\mathrm{CS}^d_{\text{prox}}} \leq \mathcal{S}[\mathrm{MQ}]$. Thus, $\mathcal{S}_{\mathrm{CS}^d_{\text{prox}}}(\mathcal{C}'_{\text{mmon}}) = m$. $\qquad \square$

**Remark 22** *Example 4 can be generalized: each concept class $\mathcal{C}$ over domain $\mathcal{X} = \{0,1\}^n$ can be (redundantly) extended to a concept class $\mathcal{C}'$ over $\mathcal{X}' = \{0,1\}^{n+1}$ that renders contrastive examples in the minimum distance model useless for learning concepts in $\mathcal{C}'$, under the Hamming distance.*

**Theorem 7** $\mathcal{S}_{\mathrm{CS}^d_{\min}}(\mathcal{C}^m_{\text{mon}} \cup \mathcal{C}^m_{\text{claus}}) = 2$, *where $d$ is the Hamming distance.*

To prove this result, we begin with a straightforward extension of Example 3.

**Lemma 23** $\mathcal{S}_{\mathrm{CS}^d_{\min}}(\mathcal{C}^m_{\text{mon}}) = 2$, *where $d$ is the Hamming distance.*

*Proof.* The proof is analogous to that of Example 3, just here the learner asks two queries, one for $\mathbf{0}$ and one for $\mathbf{1}$. The two contrastive examples then determine the target monomial, by a similar reasoning as for Example 3. $\qquad \square$

By duality[3], Lemma 23 implies the following result:

**Corollary 24** $\mathcal{S}_{\mathrm{CS}^d_{\min}}(\mathcal{C}^m_{\text{claus}}) = 2$, *where $d$ is the Hamming distance.*

We can now proceed with the proof of Theorem 7.

*Proof. (of Theorem 7.)* Let $L_\wedge$ be the learner of $\mathcal{C}^m_{\text{mon}}$ as it is described in the proof of Lemma 23. Let $L_\vee$ be the learner of $\mathcal{C}^m_{\text{claus}}$ which is obtained from $L_\wedge$ by dualization. It suffices to describe a learner $L$ who maintains simulations of $L_\wedge$ and $L_\vee$ and *either* comes in both simulations to the same

---

[3]The mapping $f(\mathbf{v}) \mapsto \bar{f}(\overline{\mathbf{v}})$ transforms the monomial $\bigwedge_{i \in I} \bar{v}_i \wedge \bigwedge_{j \in J} v_j$ into the clause $\bigvee_{i \in I} \bar{v}_i \vee \bigvee_{j \in J} v_j$, and vice versa. The same mapping can be used to dualize the proof of Lemma 23 in the sense that, within this proof, every classification label is negated and every point from $B_m$ is componentwise negated. This concerns the query points chosen by the learner as well as the contrastive examples returned by the oracle.

conclusion about the target concept[4] *or* realizes an inconsistency in one of the simulations, which can then be aborted. Details follow.

If the target concept is representable as a literal, then $L$ will not run into problems because both simulations lead to the same (correct) result. From now we assume that the target concept cannot be represented by a literal. Remember that both learners, $L_\wedge$ and $L_\vee$, choose the two query points $\mathbf{1}$ and $\mathbf{0}$. Let us first consider the case that the target function is the 1-constant (= empty monomial). Then the oracle returns the label 1 for both query points along with an error message that there is no point of opposite label in $B_m$. After having received this error message, $L$ can abort both simulations because it has identified the 1-constant as the target function. The reasoning for the constant-0 function is analogous.

Let us now consider the case that the target function can be represented by a monomial $M^*$ of length at least 2 resp. by a corresponding decision list. If $M^*$ contains at least one negated and at least one not negated variable, then the oracle returns label 0 for both query points plus an extra-point of opposite label. If the target function were a clause, then the case of label 0 for both query points can happen only if the target function would be the 0-constant (so that no extra-point of opposite label could be returned by the oracle). Hence $L$ may abort the simulation of $L_\vee$ and identify the target function via $L_\wedge$. Let us now assume that $M^*$ does not contain negated variables so that $M^*$ is of the form $\bigwedge_{j \in J} v_j$ for some $J \subseteq [m]$ with $|J| \geq 2$. Then, upon query point $\mathbf{0}$, the oracle returns label 0 plus the extra-point $\mathbf{1}_J$. The learner $L_\vee$ concludes from seeing label 0 that the target clause does not contain a negated variable. But $L_\vee$ would expect to see an extra-point with only a single 1-component (because a clause with only unnegated variables can be satisfied by a single 1-bit in the right position). Hence $L$ can abort the simulation of $L_\vee$ and continue with the simulation of $L_\wedge$ until $M^*$ is identified.

All cases that we did not discuss are dual to cases that we actually have discussed. Thus the proof of the corollary is accomplished. □

**Theorem 10** $\mathcal{S}[\mathrm{EX}^+, \mathrm{EX}^-, \mathrm{MQ}](\mathcal{C}) - 2 \leq \mathcal{S}_{\mathrm{CS}^{d_0}_{\min}}(\mathcal{C}) \leq \mathcal{S}[\mathrm{EX}^+, \mathrm{EX}^-, \mathrm{MQ}](\mathcal{C})$.

*Proof.* As for the second inequality, it suffices to show that each call of one of the oracles $\mathrm{EX}^+, \mathrm{EX}^-, \mathrm{MQ}$ can be simulated by a single call of the contrast oracle. This is clearly true for a call of the MQ-oracle (because the contrast oracle is even more informative). A call of $\mathrm{EX}^+$-oracle can be simulated as follows:

1. Choose an (arbitrary) instance $x \in \mathcal{X}$ as input of the contrast oracle and get back a pair $(b, x')$ such that $b = C^*(x) \neq C^*(x')$.

2. If $b = 1$, then return $x$ (as the desired positive example). Otherwise return $x'$.

A similar reasoning shows that a call of the $\mathrm{EX}^-$-oracle can be simulated by a call of the contrast oracle. Hence the second inequality is valid.

As for the first inequality, it suffices to show that $q$ queries of the contrast oracle can be simulated by $q$ calls of the membership oracle and two extra-calls which are addressed to the oracles providing a (positive or negative) example. For $i = 1, \ldots, q$, let $x_i$ be the $i$-th query instance that is given as input to the contrast oracle and let $(b_i, x'_i)$ with $b_i = C^*(x_i) \neq C^*(x'_i)$ be the pair returned by the latter. Note that $x'_i$ can be any instance with label opposite to $b_i$ because the discrete metric assigns the same distance value, 1, to any such pair $(x, x')$. The above $q$ queries can therefore be simulated as follows:

1. Call the oracle $\mathrm{EX}^-$ and obtain an instance $y_0$ such that $C^*(y_0) = 0$.

2. Call the oracle $\mathrm{EX}^+$ and obtain an instance $y_1$ such that $C^*(y_1) = 1$.

3. For $i = 1, \ldots, q$, choose $x_i$ as input of the MQ-oracle and get back the label $b_i = C^*(x_i)$. If $b_i = 0$, then return $(0, y_1)$ as a valid answer of the contrast oracle when the latter is called with query instance $x_i$. If $b_i = 1$, then return $(1, y_0)$ instead.

This completes the proof. □

**Example 7** *Let $\mathcal{X} = \{x_1, x_2, x_3\}$. Let $\mathcal{C}$ consist of the concepts $\{x_1, x_3\}$, $\{x_3\}$, $\{x_2, x_3\}$, and $\{x_2\}$. Then $\mathcal{C}$ satisfies $\mathrm{SD}(\mathcal{C}) = \mathrm{VCD}(\mathcal{C}) = 1$, but $\mathcal{S}_{\mathrm{CS}^d_{\min}}(\mathcal{C}) \geq 2$ for any $d : \mathcal{X} \times \mathcal{X} \to \mathbb{R}^{\geq 0}$.*

---

[4]This happens when the target concept is a literal, i.e., both a monomial of length 1 and a clause of length 1.

Table 2: A concept class $\mathcal{C}$ with $\mathrm{SD}(\mathcal{C}) = \mathrm{VCD}(\mathcal{C}) = 1$, but $\mathcal{S}_{\mathrm{CS}^d_{\min}}(\mathcal{C}) \geq 2$ for any metric $d$.

|       | $x_1$ | $x_2$ | $x_3$ |
|-------|-------|-------|-------|
| $C_1$ | 1     | 0     | 1     |
| $C_2$ | 0     | 0     | 1     |
| $C_3$ | 0     | 1     | 1     |
| $C_4$ | 0     | 1     | 0     |

*Proof.* The concept class is shown in tabular form in Table 2.

Let $d$ be any distance function and let $i(1), i(2), i(3)$ be any permutation of $1, 2, 3$. Let $E_{\{i(1),i(2)\}}$ be the event that ($L$ chooses query point $x_{i(1)}$ and $d(x_{i(2)}, x_{i(1)}) \leq d(x_{i(3)}, x_{i(1)})$) or ($L$ chooses query point $x_{i(2)}$ and $d(x_{i(1)}, x_{i(2)}) \leq d(x_{i(3)}, x_{i(2)})$). Note that at least one of the events $E_{\{1,2\}}$, $E_{\{1,3\}}$ and $E_{\{2,3\}}$ must occur. It suffices therefore to show that, in any case, the oracle has a response resulting in a version space of size 2. This can be achieved as follows:

- In case of $E_{\{1,2\}}$, the response $\{(x_1, 0), (x_2, 1)\}$ would lead to the version space $\{C_3, C_4\}$.[5]

- In case of $E_{\{1,3\}}$, the response $\{(x_1, 0), (x_3, 1)\}$ would lead to the version space $\{C_2, C_3\}$.

- In case of $E_{\{2,3\}}$, the response $\{(x_2, 0), (x_3, 1)\}$ would lead to the version space $\{C_1, C_2\}$.

$\square$

**Theorem 14** *For any finite $\mathcal{C}$ with $\mathrm{VCD}(\mathcal{C}) = 1$ there exists a metric $d$ with $\mathcal{S}_{\mathrm{CS}^d_{\min}}(\mathcal{C}) \leq 2$.*

To prove this theorem, we observe the following useful fact, which was noted by [11].

**Lemma 25 ([11])** *Let $\mathcal{C}$ be any concept class with $\mathrm{VCD}(\mathcal{C}) = 1$. Then there is at least one $(x, b) \in \mathcal{X} \times \{0,1\}$ such that there exists exactly one $C \in \mathcal{C}$ with $C(x) = b$.*

*Proof. (of Theorem 14).* It is well-known that every finite concept class of VCD 1 is contained in a concept class of size $|\mathcal{X}| + 1$ whose VCD is still 1, see, e.g., [12]. Thus, we may assume that $\mathcal{C} = |\mathcal{X}| + 1$. Set $n = |\mathcal{X}|$. By an iterative application of Lemma 25, we see that there exists an order $C_1, \ldots, C_n, C_{n+1}$ of the concepts in $\mathcal{C}$, an order $x_1, \ldots, x_n$ of the instances in $\mathcal{X}$ and a binary sequence $b_1, \ldots, b_n$ such that, for $i = 1, \ldots, n$, $C = C_i$ is the only concept in $\mathcal{C} \setminus \{C_1, \ldots, C_{i-1}\}$ which satisfies the equation $C(x_i) = b_i$. Let $i^+(1) < \ldots < i^+(r)$ and $i^-(1) < \ldots < i^-(s)$ with $r + s = n$ be given by $b_{i^+(j)} = 1$ for $j = 1, \ldots, r$ and $b_{i^-(j)} = 0$ for $j = 1, \ldots, s$. In the sequel, we assume that $r, s \geq 1$.[6] The following implications are easy to verify.

**(I1)** If $C(x_j) = 1 - b_j$ for all $j \in [n]$, then $C = C_{n+1}$.

**(I2)** If $C(x_{i^+(j)}) = 0$ for all $j \in [r]$, and $j^-$ is the smallest $j \in [s]$ with $C(x_{i^-(j)}) = 0$, then $C = C_{i^-(j^-)}$.

**(I3)** If $C(x_{i^-(j)}) = 1$ for all $j \in [s]$, and $j^+$ is the smallest $j \in [r]$ with $C(x_{i^+(j)}) = 1$, then $C = C_{i^+(j^+)}$.

**(I4)** If (i) $j^-$ is the smallest $j \in [s]$ such that $C(x_{i^-(j)}) = 0$, (ii) $j^+$ is the smallest $j \in [r]$ such that $C(x_{i^+(j)}) = 1$, and (iii) $i^+(j^+) > i^-(j^-)$, then $C = C_{i^-(j^-)}$.

**(I5)** If (i) $j^-$ is the smallest $j \in [s]$ such that $C(x_{i^-(j)}) = 0$, (ii) $j^+$ is the smallest $j \in [r]$ such that $C(x_{i^+(j)}) = 1$, and (iii) $i^-(j^-) > i^+(j^+)$, then $C = C_{i^+(j^+)}$.

It thus suffices to choose a metric $d$ such that the oracle can be forced (with 2 minimum-distance queries) to make one of the above five implications applicable. To this end, we define

$$d(x_i, x_j) = \begin{cases} |i - j| & \text{if } b_i = b_j \\ n & \text{if } b_i \neq b_j \end{cases}.$$

---

[5]The learner can take no advantage from knowing which of $x_1, x_2$ is in the role of the query point and which is in the role of the contrastive example.

[6]The proof will become simpler if $r$ or $s$ equals 0.

The two query points, chosen by the learner, are $x_{i^+(1)}$ and $x_{i^-(1)}$.

Suppose first that the oracle returns the label 1 for $x_{i^+(1)}$. If $i^+(1) = 1$, then $C = C_1$ and we are done. We may therefore assume that $i^+(1) > 1$, which implies that $i^-(1) = 1$. If the oracle returns the label 0 for $x_{i^-(1)} = x_1$, then $C = C_1$ and we are done. We may therefore assume that the oracle returns the label 1 for $x_1$. If there is no contrastive example for $(x_1, 1)$, then (I3) applies. Otherwise, let $(x_j, 0)$ be the contrastive example for $(x_1, 1)$. If $j < i^+(1)$, then (I4) applies. If $j > i^+(1)$, then either (I3) (in case $d(x_1, x_j) = n$) or (I5) (in case $d(x_1, x_j) < n$) applies.

Due to symmetry, one can reason analogously if the oracle returns the label 0 for $x_{i^-(1)}$.

Finally, consider the case that the oracle returns the label 0 for $x_{i^+(1)}$ and the label 1 for $x_{i^-(1)}$. It follows that the contrastive example for $x_{i^+(1)}$ must be labeled 1 and the contrastive example for $x_{i^-(1)}$ must be labeled 0. The following implications are easy to verify:

1. If the contrastive example for $x_{i^+(1)}$ is an instance $x_i$ with $b_i = 0$ and the contrastive example for $x_{i^-(1)}$ is an instance $x_i$ with $b_i = 1$, then (I1) applies.

2. If the contrastive example for $x_{i^+(1)}$ is an instance $x_i$ with $b_i = 0$ and the contrastive example for $x_{i^-(1)}$ is an instance of the form $x_{i^-(j^-)}$, then (I2) applies.

3. If the contrastive example for $x_{i^-(1)}$ is an instance $x_i$ with $b_i = 1$ and the contrastive example for $x_{i^+(1)}$ is an instance of the form $x_{i^+(j^+)}$, then (I3) applies.

4. If (i) the contrastive example for $x_{i^+(1)}$ is an instance of the form $x_{i^+(j^+)}$, (ii) the contrastive example for $x_{i^-(1)}$ is an instance of the form $x_{i^-(j^-)}$, and (iii) $i^+(j^+) > i^-(j^-)$, then (I4) applies.

5. If (i) the contrastive example for $x_{i^+(1)}$ is an instance of the form $x_{i^+(j^+)}$, (ii) the contrastive example for $x_{i^-(1)}$ is an instance of the form $x_{i^-(j^-)}$, and (iii) $i^-(j^-) > i^+(j^+)$, then (I5) applies.

Hence each possible response of the oracle leads to the exact identification of the target concept. $\square$

**Lemma 15** *Suppose that $\mathcal{C}$ is a concept class over $B_m$, and $\mathcal{C}'$ is a subclass of $\mathcal{C}$ with the properties*
*(P1) For each $\mathbf{a} \in B_m$ with $a_m = y_m$ and each $C \in \mathcal{C}'$, we have that $C(\mathbf{a}) = 1$.*
*(P2) For each $\mathbf{a}$ with $a_m = y'_m$ and $a_{m-1} = y_{m-1}$ and each $C \in \mathcal{C}'$, we have that $C(\mathbf{a}) = 0$.*
*For $i \in \{m, m-1\}$, here $y_i \in \{0, 1\}$ is fixed, $y'_i = 1 - y_i$, and $\mathcal{C}''$ is the set of all concepts of shape*

$$(a_1, \ldots, a_{m-2}) \mapsto C(a_1, \ldots, a_{m-2}, y'_{m-1}, y'_m) \text{ for } C \in \mathcal{C}'.$$

*Then $\mathcal{S}_{\mathrm{CS}^d_{\min}}(\mathcal{C}) \geq \mathcal{S}_{\mathrm{CS}^d_{\min}}(\mathcal{C}') \geq \mathcal{S}[\mathrm{MQ}](\mathcal{C}'')$, where $d$ is the Hamming distance.*

*Proof.* Suppose that $L$ learns $\mathcal{C}'$ from a contrast oracle in the minimum distance model at the expense of $q$ queries. It suffices to show that $L$ can be transformed into $L'$ which learns $\mathcal{C}''$ from a membership oracle at the expense of at most $q$ queries. To this end, let $C''$ be the target concept in $\mathcal{C}''$ and let $C'$ with $C'(\mathbf{a}y'_{m-1}y'_m) = C''(\mathbf{a})$ be the corresponding concept in $\mathcal{C}'$. Note that $L$ has not uniquely identified $C'$ in $\mathcal{C}'$ as long as the subfunction $C''$ is not uniquely identified in $\mathcal{C}''$. $L'$ can therefore uniquely identify $C''$ in $\mathcal{C}''$ by maintaining a simulation of $L$ until $L$ has uniquely identified $C'$ in $\mathcal{C}$. In order to explain how $L'$ simulates the contrast oracle, we proceed by case distinction:

- If $L$ chooses a query point of the form $\mathbf{a}y'_{m-1}y'_m$, then $L'$ chooses the query point $\mathbf{a}$ and receives the label $b := C''(\mathbf{a})$ from its membership oracle. Then $L'$ returns $b$ and the point $\mathbf{a}'$ to $L$ where $\mathbf{a}' = \mathbf{a}y'_{m-1}y_m$ if $b = 0$ and $\mathbf{a}' = \mathbf{a}y_{m-1}y'_m$ if $b = 1$. Note that $(b, \mathbf{a}')$ is among the admissible answers of the contrast oracle.

- If $L$ chooses a query point of the form $\mathbf{a}y_{m-1}y'_m$, then $L'$ returns the label 0 and the additional point $\mathbf{a}y_{m-1}y_m$. Again, this is among the admissible answers of the contrast oracle.

- If $L$ chooses a query point of the form $\mathbf{a}y_{m-1}y_m$, then $L'$ returns the label 1 and the additional point $\mathbf{a}y_{m-1}y'_m$. Again, this is among the admissible answers of the contrast oracle.

- If $L$ chooses a query-point of the form $\mathbf{a}y'_{m-1}y_m$, then $L'$ chooses the query point $\mathbf{a}$ and receives the label $b = C''(\mathbf{a})$ from its membership oracle. Then $L'$ returns the label 1 and

the point $\mathbf{a}'$ to $L$ where $\mathbf{a}' = \mathbf{a}y'_{m-1}y'_m$ if $b = 0$ and $\mathbf{a}' = \mathbf{a}y_{m-1}y'_m$ if $b = 1$. Again, this is among the admissible answers of the contrast oracle. Here note that, in case $b = 1$, there is no admissible answer at distance 1 from the query point, so that a contrastive example at distance 2 can be chosen.

Clearly $L''$ can maintain this simulation until it reaches exact identification of the target concept. Moreover, the number of query point chosen by $L'$ does not exceed the number of query points chosen by $L$. □

**Corollary 16** $\mathcal{S}_{\mathrm{CS}^d_{\min}}(\mathrm{DL}^2_m) \geq 2^{m-2} - 1$, *where $d$ is the Hamming distance.*

*Proof.* Let $\mathcal{C} = \mathrm{DL}^2_m$. Moreover, let $\mathcal{C}' \subseteq \mathcal{C}$ be a subset of $\mathrm{DL}^2_m$ of decision lists of the form

$$\mathcal{L} = [(v_m, 1), (v_{m-1}, 0), \underbrace{(\ell_1, b_1), \dots (\ell_z, b_z), b_{z+1}}_{=\mathcal{L}' \in \mathrm{DL}^1_{m-2}}] \in \mathrm{DL}^2_m \ ,$$

where $z \geq 0$, $\ell_i \in \{v_1, \bar{v}_1, \dots, v_{m-2}, \bar{v}_{m-2}\}$ and $b_i \in \{0, 1\}$. $\mathcal{C}'$ satisfies properties (P1) and (P2) with $y_{m-1} = y_m = 1$ in Lemma 15. Then $\mathcal{C}'' = \mathrm{DL}^1_{m-2}$. $\mathcal{C}^{m-2}_{\mathrm{mon}}$ is a subclass of $\mathrm{DL}^1_{m-2}$ and $2^{m-2} - 1$ membership queries are needed for learning $\mathcal{C}^{m-2}_{\mathrm{mon}}$ [17]. □

**Theorem 17** *There exists a self-directed learner for $\mathrm{DL}_m$ that makes, on any target list $C^* \in \mathrm{DL}_m$, at most $4km$ mistakes where $k$ is the number of blocks in $C^*$.*

*Proof.* Recall $B_m = \{0, 1\}^m$. For any set of literals $\ell_{i_1}, \dots, \ell_{i_j}$, we use $B_m(\ell_{i_1} = b_1, \dots, \ell_{i_j} = b_j)$ to denote the $(m - j)$-dimensional subcube resulting from fixing the value of $\ell_{i_t}$ to $b_t$, for $t \in [j]$. We call a literal $\ell$ $b$-pure if the target list $C^*$ assigns label $b$ to every point in $B_m(\ell = 1)$.

The $b$-pureness of $\ell = v_i$ (resp. of $\ell = \bar{v}_i$) can be checked by testing whether $C^*$ restricted to $B_m(v_i = 1)$ (resp. to $B_m(v_i = 0)$) degenerates to the constant-$b$ function. This is done at the expense of at most 2 mistakes per literal: The learner predicts 0 for a first point in $B_m(\ell = 1)$, receives the correct label $b$ (which may or may not count as one mistake), and then predicts $b$ for further points in $B_m(\ell = 1)$ until a mistake is made. If no mistake is made on further points in $B_m(\ell = 1)$, then $\ell$ is $b$-pure; otherwise it is not pure. Hence, after having made at most $4m$ mistakes, the learner $L$ knows all pure-literals, say $\ell_1, \dots, \ell_t$, and $L$ also knows the (unique) bit $b \in \{0, 1\}$ with respect to which $\ell_1, \dots, \ell_t$ are pure.

At this point $L$ knows the first block $(\ell_1, b), \dots (\ell_t, b)$ of the list $C^*$. $L$ can now proceed iteratively in order to learn the remaining blocks. As for the second block, all query points are taken from the $(m - t)$-dimensional subcube $B_m(\ell_1 = 1 - b, \dots, \ell_t = 1 - b)$ so that they are not absorbed by the first block. In this way, $C^*$ is identified blockwise at the expense of at most $4m$ queries per block. □

**Theorem 21** $\mathcal{S}_{CS^{(d_{\mathcal{C}_t})}_{\min}}(\mathcal{C}^{m,s,z}_{\mathrm{MDNF}}) \leq s = \mathrm{SD}(\mathcal{C}^{m,s,z}_{\mathrm{MDNF}})$, *where $\mathcal{C}_t = (\mathcal{C}^{m,s,z}_{\mathrm{MDNF}})_t$ for any $t$.*

The proof of this theorem makes use of the following terminology. Suppose $\mathbf{a} = (a_1, \dots, a_m)$, $\mathbf{b} = (b_1, \dots, b_m) \in B^m$ for some $m$. We write $\mathbf{a} \leq \mathbf{b}$ if $a_i = 1$ implies $b_i = 1$. Moreover, we implicitly identify the Boolean vector $\mathbf{a}$ with the corresponding monotone monomial.

First we introduce a helpful lemma.

**Lemma 26** *Let $\mathbf{a}, \mathbf{b}, \mathbf{c} \in \{0, 1\}^m$ such that $\mathbf{a} \leq \mathbf{b} \leq \mathbf{c}$. Then for any $\mathcal{C}' \subseteq \mathcal{C}^{m,s,z}_{\mathrm{MDNF}}$ we have $d_{\mathcal{C}'}(\mathbf{a}, \mathbf{b}) \leq d_{\mathcal{C}'}(\mathbf{a}, \mathbf{c})$.*

*Proof.* Consider any $C \in \mathcal{C}^{m,s,z}_{\mathrm{MDNF}}$. Since $C$ is monotone, if $C(\mathbf{a}) = 1$ then we have $C(\mathbf{b}) = 1, C(\mathbf{c}) = 1$. Similarly, if $C(\mathbf{a}) = 0$ and $C(\mathbf{b}) = 1$ then $C(\mathbf{c}) = 1$ as well. This completes the proof. □

*Proof. (of Theorem 21).* The claim $\mathrm{SD}(\mathcal{C}^{m,s,z}_{\mathrm{MDNF}}) = s$ was proven by [13] and already stated in Theorem 19. It suffices to define a learner that witnesses $\mathcal{S}_{CS^{(d_{\mathcal{C}_t})}_{\min}}(\mathcal{C}^{m,s,z}_{\mathrm{MDNF}}) \leq s$.

Define a learner that asks, in each of $s$ iterations, a query for the vector $\mathbf{0}$. Since $\mathbf{0}$ has label 0, all the contrastive examples have label 1. Let $\mathbf{b}^t$ be the contrastive example for the $t$th query.

First, we argue that $\mathbf{b}^1$ is a monomial in the target concept $C^*$. Suppose $\mathbf{b}^1$ is not a monomial. Since it has label 1, there is a monomial $\mathbf{b}' \le \mathbf{b}^1$. Using Lemma 26, we have $d_{\mathcal{C}_1}(0, \mathbf{b}') \le d_{\mathcal{C}_1}(0, \mathbf{b}^1)$. Moreover, since $\mathbf{b}^1 \ne \mathbf{b}'$, there exists $C \in \mathcal{C}_{\mathrm{MDNF}}^{m,s,z}$ such that $C(\mathbf{b}') = 0$, but $C(\mathbf{b}^1) = 1$. Thus, $d_{\mathcal{C}_1}(0, \mathbf{b}') < d_{\mathcal{C}_1}(0, \mathbf{b}^1)$. Therefore $\mathbf{b}'$ should have been the contrastive example, which is a contradiction.

Next, we prove inductively that $\{\mathbf{b}^1, ..., \mathbf{b}^s\}$ is the set of monomials in $C^*$. Let $\{\mathbf{b}^1, ..., \mathbf{b}^{t-1}\}$ be $t - 1$ distinct monomials in $C^*$. We prove that $\mathbf{b}^t$ is a monomial in $C^*$ distinct from $\mathbf{b}^1, ..., \mathbf{b}^{t-1}$.

Note that $\mathcal{C}_{t-1}$ is the set of concepts that have label 1 on all of $\mathbf{b}^1, ..., \mathbf{b}^{t-1}$. We also know that all concepts give label 0 to the all zero vector. Thus, according to Lemma 26 for any $\mathbf{a}$ with $\mathbf{b}^q \le \mathbf{a}$ for some $q \in [t - 1]$, we have $d_{\mathcal{C}_1}(0, \mathbf{a}) \ge d_{\mathcal{C}_1}(0, \mathbf{b}^q) = 1$ (maximum distance). Thus, $\mathbf{b}^q \not\le \mathbf{b}^t$. With an argument similar to one we used for the base case, $\mathbf{b}^t$ is also a monomial. This completes the proof. $\qquad\square$

