# OpenReview forum: "Formal Models of Active Learning from Contrastive Examples"
_NeurIPS.cc/2025/Conference — NeurIPS 2025 poster_

### Official Review · Reviewer_W69F · 2025-06-22

**Clarity:** 3
**Significance:** 2
**Originality:** 3
**Rating:** 5
**Confidence:** 3

**Summary:**

This paper considers theoretical upper and lower bounds of label complexity for a variant of active learning, where the oracle can give a "contrastive" example in addition to the label of the query example. It considers two specific types or oracles for contrastive examples: (1) $CS_{min}$: it always gives the example that is closest to the query example but has a different label; (2) $CS_{prox}$: it gives any example whose distance to the query example is less than $r$ and whose label is different. It assumes that the learner knows the oracle's algorithm for finding contrastive examples, and that the oracle always gives the correct label for the query example.

It mainly considers two types of concept classes: thresholds/rectangales, and variants of monotone Boolean functions, and consequently the distance metric for the contrastive oracle is l1 and hamming distance. For k-dimensional rectangles, it shows that with $CS_{prox}$ can improve the label complexity by a factor of k, and for $CS_{min}$ (which is quite strong), it improves label complexity to O(1). For Boolean functions, it shows that both $CS_{prox}$ and $CS_{min}$ reduces label complexity for some simple variants of monotone monomials, but not for others. In terms of lower bound, it shows a nice reduction from self-directed learning (to some degree a mix of online and active learning), and thus gives a metric-agnostic lower bound based on the mistake bound of self-direct learning.  It also gives some lower bounds  for $CS_{min}$ depending on the hamming distance.

**Questions:**

- Regarding significance, I would appreciate if the authors could comment on the limitation I mentioned in the Strengths And Weaknesses section.

- (Minor) For the lower bound in section 4.1., the connection to Self-directed learning looks nice, but how strong is this result? Or is there any interesting insights from it?

**Ethical Concerns:**

["NO or VERY MINOR ethics concerns only"]

**Final Justification:**

The author response addresses my issues: the limitations remain valid, but the results obtained are nontrivial and interesting enough. I would recommend this paper to be accepted.

**Limitations:**

yes

**Paper Formatting Concerns:**

None.

**Quality:**

3

**Strengths And Weaknesses:**

Quality: The results look technically correct and sound to me (I quickly went through the proof sketches in the main text and did not check proofs in Appendix).

Clarity: The paper is overall well organized and written. It is pretty definition and theory heavy, but it provides plenty of explanation, examples, and proof sketches, which is helpful.

Significance: This paper considers an interesting and relatively new theoretical problem for active learning. On the downside, the setting and assumption of the paper is quite restrictive:  for concept class, only simple thresholds and boolean functions are considered; for oracle, it assumes it is noiseless, and only considers two simple "contrastive" types (min and prox). This limits its impact to both theory and practical community.

Originality: I'm not very familiar with the literature on contrastive learning, but it looks to me the techniques and results are novel, and related work is sufficiently cited and discussed.

---

> ### Author Rebuttal · Authors · 2025-07-31
>
> # General Response to All Reviewers
> Thank you for your effort in assessing our work, and for your helpful comments and questions. In this section, we address a common concern raised by multiple reviewers: that our setting is overly idealistic, as it assumes a realizable setup (i.e., the target concept lies within the concept class) and that the learner has perfect knowledge of the oracle.
>
> While we agree that extending the framework to agnostic settings and scenarios involving noise or imperfect alignment between learner and oracle is an important direction for future work, we argue that the current setup still offers meaningful contributions to the literature:
> - The primary aim of this paper is to initiate a theoretical study of learning with contrastive examples. As a first step, it is natural and appropriate to begin by exploring the boundaries of idealized, tractable setups.
> - We establish that, for any distance metric, the complexity of the minimum distance model is lower bounded by the complexity of self-directed learning (a topic that has recently re-gained attention in the learning theory community). To the best of our knowledge, there are few known concept classes where self-directed learning outperforms active learning in terms of sample complexity, with the notable exception of the result by Diakonikolas et al. (2023) for halfspaces over the uniform distribution. Thus, any upper bounds from our idealized setup carry over directly to self-directed learning.
> - As we highlighted in the paper, one of the most prominent applications of our model is counterexample-guided program synthesis, where it is in many cases guaranteed that the learner has perfect knowledge of the oracle and the target concept lies in the concept class.
>
> In addition, we would like to point out that our framework naturally extends to setups where the learner has imperfect knowledge of how the oracle selects contrastive examples; see also our response to Reviewer 9itQ.
>
> # Individual Response
> We would like to address your questions bellow:
> 1. While we agree with most of the limitations raised by the reviewer—and have addressed them in our general response to all reviewers—we would like to emphasize that the concept classes introduced in Section 4, although composed of Boolean functions, are non-trivial to learn.
> 2. Importantly, any lower bound established for self-directed learning directly applies to the minimum-distance model. Furthermore, self-directed learning has recently attracted renewed attention in the learning theory community (see, e.g., Devulapalli and Hanneke, 2024).
>
> References: \
> Diakonikolas, Ilias, et al. "Self-directed linear classification." The Thirty Sixth Annual Conference on Learning Theory. PMLR, 2023.\
> Devulapalli, Pramith, and Steve Hanneke. "The dimension of self-directed learning." International Conference on Algorithmic Learning Theory. PMLR, 2024.

---

### Official Review · Reviewer_brPw · 2025-07-03

**Clarity:** 3
**Significance:** 2
**Originality:** 3
**Rating:** 4
**Confidence:** 2

**Summary:**

The authors introduce a formal model for learning from contrastive examples and study contrast sample complexity for several examples. Learning goes according to the following protocol. A learner makes a query $x_i \in \mathcal X$. After that, a contrast oracle returns a triplet $[y_i, (x_i', y_i')]$, where $y_i$ and $y_i'$ are the labels of $x_i$ and $x_i'$, respectively and $x_i'$ belongs to the contrast set of the current version space and $x_i$. Then the learner reduces the version space and makes a new query. The authors start with an observation that they have to restrict the power of the contrast oracle (Proposition 2). Otherwise, the learning problem will be too simple. In the present submission, the authors consider two contrast oracles. The first one returns the closest element $x_i'$ with a different label while the second one is allowed to take any contrastive example $x_i'$ within a ball of radius $r$ around $x_i$. The authors derive contrast sample complexity for classes of thresholds (Example 1), axis-aligned rectangles (Example 2), monotone monomials (Example 3), and the union of the class of monomials of Boolean variables and the dual class of clauses (Theorem 7). In Example 4, the authors describe a simple scenario when the contrast oracle may be useless. Finally, they discuss connection between learning from contrastive examples and the classical model of self-directed learning.

**Questions:**

1. What happens to the contrast sample complexity if the target concept lies outside $\mathcal C$?

2. Is it possible extend the considered model to scenarios when labels are noisy?

3. Can the authors give an example of a practical task when a contrast oracle knows the contrast class exactly?

**Ethical Concerns:**

["NO or VERY MINOR ethics concerns only"]

**Final Justification:**

I thank the authors for their clarifications. I still think that the setup is too idealistic. However, in my initial review I underestimated the connection with self-directed learning. For this reason, I raised my score.

**Limitations:**

Yes.

**Paper Formatting Concerns:**

No.

**Quality:**

3

**Strengths And Weaknesses:**

Strengths.

1. The paper introduces formal models of contrastive learning and provides its theoretical foundations.

2. The authors provide several examples of contrast sample complexity for concept classes often considered in the literature on learning theory: thresholds, axis-aligned rectangles, monotone monomials.

3. The authors discuss connection between learning from contrastive examples and the model of self-directed learning.


Weaknesses.

1. The contrast sample complexity in several examples of order $\mathcal O(1)$. This means that the problem of contrastive learning in these cases is too simple.

2. The setup is too ideal. First, the oracle has an access to exact contrastive set. Second, the labels are noiseless. Third, the target concept $C^*$ belongs to the concept class $\mathcal C$.

---

> ### Author Rebuttal · Authors · 2025-07-31
>
> # General Response to All Reviewers
> Thank you for your effort in assessing our work, and for your helpful comments and questions. In this section, we address a common concern raised by multiple reviewers: that our setting is overly idealistic, as it assumes a realizable setup (i.e., the target concept lies within the concept class) and that the learner has perfect knowledge of the oracle.
>
> While we agree that extending the framework to agnostic settings and scenarios involving noise or imperfect alignment between learner and oracle is an important direction for future work, we argue that the current setup still offers meaningful contributions to the literature:
> - The primary aim of this paper is to initiate a theoretical study of learning with contrastive examples. As a first step, it is natural and appropriate to begin by exploring the boundaries of idealized, tractable setups.
> - We establish that, for any distance metric, the complexity of the minimum distance model is lower bounded by the complexity of self-directed learning (a topic that has recently re-gained attention in the learning theory community). To the best of our knowledge, there are few known concept classes where self-directed learning outperforms active learning in terms of sample complexity, with the notable exception of the result by Diakonikolas et al. (2023) for halfspaces over the uniform distribution. Thus, any upper bounds from our idealized setup carry over directly to self-directed learning.
> - As we highlighted in the paper, one of the most prominent applications of our model is counterexample-guided program synthesis, where it is in many cases guaranteed that the learner has perfect knowledge of the oracle and the target concept lies in the concept class.
>
> In addition, we would like to point out that our framework naturally extends to setups where the learner has imperfect knowledge of how the oracle selects contrastive examples; see also our response to Reviewer 9itQ.
>
> # Individual Response
> We have tried to address the questions you raised in our general response to all reviewers.
>
>
> References:
>
> Diakonikolas, Ilias, et al. "Self-directed linear classification." The Thirty Sixth Annual Conference on Learning Theory. PMLR, 2023.

---

> > ### Comment · Reviewer_brPw · 2025-08-02
> >
> > Thank you for the answers. I have no additional questions.

---

### Official Review · Reviewer_MWJZ · 2025-07-03

**Clarity:** 3
**Significance:** 3
**Originality:** 4
**Rating:** 5
**Confidence:** 4

**Summary:**

Formal Models of Active Learning from Contrastive Examples
This dense theoretical paper proposes a framework to analyze active learning where a learner, in addition to a label for a query, also receives a "contrastive example" which is a similar example with a different label. The authors assume such examples are drawn in a couple of different ways, using distances from the query. They show sample complexity results for different hypothesis classes, relationships to the SQ and self directed learning. In some cases, the additional contrastive examples have a significant impact on sample complexity, in other cases not so much.

**Questions:**

see Strengths and Weaknesses above

**Ethical Concerns:**

["NO or VERY MINOR ethics concerns only"]

**Limitations:**

yes

**Quality:**

3

**Strengths And Weaknesses:**

This paper is addressing an important and relevant problem. Active learning is an important research area and studying the sample complexity under contrastive examples is an interesting direction. In spite of the density of the paper, it is very well written and explained. Generally I need a linear (in the number of lemmas) number of cups of coffee to read such papers. For this paper, three cups were enough to gain some understanding of 21 theorems. That is a significant improvement!

A few concerns:
1. Such examples are called counterfactuals (l 19)--I don't understand. Is there an underlying causal model behind these choices? In what sense are they counterfactual?
2. I was not clear on the value of the contrastive example per se. As in, what if at each round a random example was provided  with the opposite label? How much worse would the sample complexities be? Maybe this was addressed and I missed it.
3. I believe there is work on learning with counterexamples, how does this compare to such a framework?
4. L is overused as a symbol. Maybe lists of oracles should be called something else.
5. What does it mean to say "access to all oracles"? The learner gets multiple examples one from each?
6. What is the significance of x_{i+1} in definition 1? Why is it not enough to say sigma epsilon-approximates C*?

On the whole, this is an interesting, well written paper studying a novel setting theoretically. These results should be of interest to the community.

---

> ### Author Rebuttal · Authors · 2025-07-31
>
> # General Response to All Reviewers
> Thank you for your effort in assessing our work, and for your helpful comments and questions. In this section, we address a common concern raised by multiple reviewers: that our setting is overly idealistic, as it assumes a realizable setup (i.e., the target concept lies within the concept class) and that the learner has perfect knowledge of the oracle.
>
> While we agree that extending the framework to agnostic settings and scenarios involving noise or imperfect alignment between learner and oracle is an important direction for future work, we argue that the current setup still offers meaningful contributions to the literature:
> - The primary aim of this paper is to initiate a theoretical study of learning with contrastive examples. As a first step, it is natural and appropriate to begin by exploring the boundaries of idealized, tractable setups.
> - We establish that, for any distance metric, the complexity of the minimum distance model is lower bounded by the complexity of self-directed learning (a topic that has recently re-gained attention in the learning theory community). To the best of our knowledge, there are few known concept classes where self-directed learning outperforms active learning in terms of sample complexity, with the notable exception of the result by Diakonikolas et al. (2023) for halfspaces over the uniform distribution. Thus, any upper bounds from our idealized setup carry over directly to self-directed learning.
> - As we highlighted in the paper, one of the most prominent applications of our model is counterexample-guided program synthesis, where it is in many cases guaranteed that the learner has perfect knowledge of the oracle and the target concept lies in the concept class.
>
> In addition, we would like to point out that our framework naturally extends to setups where the learner has imperfect knowledge of how the oracle selects contrastive examples; see also our response to Reviewer 9itQ.
>
> # Individual Response
> We would like to address your questions below:
> 1. The term counterfactual is often used to describe feedback provided in response to a query to an oracle, where the feedback contrasts with the query in some specific aspect. In this sense, there are clear similarities to our notion of a contrastive example, which has an opposite label with the original instance.
> 2. One can consider the class of 1-dimensional threshold functions (Example 1 of the paper). The minimum distance model contrastive examples identifies the target concept with 1 example. However, it is not hard to see that with a random contrastive example with opposite labels the sample complexity will remain $\Theta( \log(1 / \varepsilon))$ (the same as the membership query complexity).
> 3. Are you referring to Alon et al. (2023)? If so, their setup is closer to representation learning than to ours. In their framework, the learner receives triplets $(x, y, z)$ and learns whether $x$ is closer to $y$ or to $z$. The goal is to correctly predict the outcome for all such triplet comparisons. They also extend this framework to more general $k$-tuple queries. \
> If you are referring to counterexamples used in Angluin-style learning with equivalence queries or superset/subset queries: here the queries are not data points but entire concepts (hypotheses), and the counterexamples are used to demonstrate why the hypothesis is incorrect.
> 4. We will address this problem in the camera ready version, if this paper gets accepted.
> 5. By access to all oracles, we mean that at each step $t$, the learner not only selects a query point $x_t$ but also chooses which oracle to query for a contrastive example. We will clarify this point in the paper.
> 6. In line 99, within Definition 1, our intention was to emphasize that the learner selects each $x_{i+1}$​ based on the full interaction history observed up to that point.
>
> References: \
> Diakonikolas, Ilias, et al. "Self-directed linear classification." The Thirty Sixth Annual Conference on Learning Theory. PMLR, 2023. \
> Alon, Noga, et al. "Optimal sample complexity of contrastive learning." arXiv preprint arXiv:2312.00379 (2023).

---

### Official Review · Reviewer_C2ZM · 2025-07-05

**Clarity:** 3
**Significance:** 3
**Originality:** 3
**Rating:** 5
**Confidence:** 4

**Summary:**

This paper introduces a formal framework for learning from contrastive examples, focusing on scenarios where the learner has prior knowledge about the oracle's strategy for selecting such examples. The first contribution is the development of a general and flexible framework that captures a variety of contrastive example generation methods, extending beyond traditional counterfactual settings. The second contribution is providing various sample complexities within this framework. The authors derive both upper and lower bounds for learning different concept classes, including geometric objects, 1-decision lists, and monotone DNFs. These results show implications of contrastive feedback on learning efficiency. The third contribution is an interesting connection between the sample complexity of contrastive learning and the mistake bound in self-directed online learning.

**Questions:**

I think the paper is super clear.

**Ethical Concerns:**

["NO or VERY MINOR ethics concerns only"]

**Limitations:**

I think the major limitation of this framework is that it allows the learner to reason about the choice of the contrast set with perfect knowledge. I understand why this assumption is made in order to make the proofs work. However, it has to be relaxed for it to be really informative about what happens in practice.

**Quality:**

3

**Strengths And Weaknesses:**

I really like this paper. It makes very neat theoretical contributions (in a field that is mostly empirical). So in my opinion, the paper should be accepted.

---

> ### Author Rebuttal · Authors · 2025-07-31
>
> # General Response to All Reviewers
> Thank you for your effort in assessing our work, and for your helpful comments and questions. In this section, we address a common concern raised by multiple reviewers: that our setting is overly idealistic, as it assumes a realizable setup (i.e., the target concept lies within the concept class) and that the learner has perfect knowledge of the oracle.
>
> While we agree that extending the framework to agnostic settings and scenarios involving noise or imperfect alignment between learner and oracle is an important direction for future work, we argue that the current setup still offers meaningful contributions to the literature:
> - The primary aim of this paper is to initiate a theoretical study of learning with contrastive examples. As a first step, it is natural and appropriate to begin by exploring the boundaries of idealized, tractable setups.
> - We establish that, for any distance metric, the complexity of the minimum distance model is lower bounded by the complexity of self-directed learning (a topic that has recently re-gained attention in the learning theory community). To the best of our knowledge, there are few known concept classes where self-directed learning outperforms active learning in terms of sample complexity, with the notable exception of the result by Diakonikolas et al. (2023) for halfspaces over the uniform distribution. Thus, any upper bounds from our idealized setup carry over directly to self-directed learning.
> - As we highlighted in the paper, one of the most prominent applications of our model is counterexample-guided program synthesis, where it is in many cases guaranteed that the learner has perfect knowledge of the oracle and the target concept lies in the concept class.
>
> In addition, we would like to point out that our framework naturally extends to setups where the learner has imperfect knowledge of how the oracle selects contrastive examples; see also our response to Reviewer 9itQ.
>
> # Individual Response
> We have tried to address the limitations you raised in our general response to all reviewers.
>
>
> References:
>
> Diakonikolas, Ilias, et al. "Self-directed linear classification." The Thirty Sixth Annual Conference on Learning Theory. PMLR, 2023.

---

### Official Review · Reviewer_9itQ · 2025-07-10

**Clarity:** 3
**Significance:** 3
**Originality:** 4
**Rating:** 4
**Confidence:** 1

**Summary:**

This paper proposes a general framework for active learning in which a membership query returns not only a label but also a counter-example chosen according to a rule known to the learner. The authors analyze two specific rules - a strict minimum-distance rule (the closest counter-example to the query with a different label) and a looser proximity rule (any counter-example within a certain neighborhood). They show that for simple concept classes such as one-dimensional thresholds and axis-aligned rectangles the extra feedback dramatically reduces the number of queries to a constant. Finally, they link the framework to self-directed learning, allowing them to import existing lower-bound techniques and to derive a new mistake bound for decision lists.

**Questions:**

How does the sample complexity change if the learner's assumed metric differs slightly from the oracle's, or if the oracle occasionally returns a non-minimal counter-example? A brief discussion would help gauge practical relevance.

**Ethical Concerns:**

["NO or VERY MINOR ethics concerns only"]

**Final Justification:**

The rebuttal cleared up my concerns about the idealized setting and made the scope feel reasonable for a first theoretical step. I will maintain my Borderline Accept (4) rating.

**Limitations:**

Yes.

**Paper Formatting Concerns:**

None.

**Quality:**

4

**Strengths And Weaknesses:**

The paper's main merit is a new framework that can unify several earlier "teacher" settings - greedy, randomized, and adversarial - giving one vocabulary what used to be separate analyses. Coupled with complete proofs, the structural unification could facilitate future research in this area.

The work leans on a strong perfect-knowledge assumption - the learner must know the oracle's exact rule and metric - yet offers almost no guidance on robustness to noise or mis-specification. All evidence remains theoretical; without even a toy simulation it is unclear whether the constant-query gains would survive in practice. Moreover, the most striking positive results (thresholds and rectangles) are arguably obvious, leaving doubts about practical impact.

---

> ### Author Rebuttal · Authors · 2025-07-31
>
> # General Response to All Reviewers
> Thank you for your effort in assessing our work, and for your helpful comments and questions. In this section, we address a common concern raised by multiple reviewers: that our setting is overly idealistic, as it assumes a realizable setup (i.e., the target concept lies within the concept class) and that the learner has perfect knowledge of the oracle.
>
> While we agree that extending the framework to agnostic settings and scenarios involving noise or imperfect alignment between learner and oracle is an important direction for future work, we argue that the current setup still offers meaningful contributions to the literature:
> - The primary aim of this paper is to initiate a theoretical study of learning with contrastive examples. As a first step, it is natural and appropriate to begin by exploring the boundaries of idealized, tractable setups.
> - We establish that, for any distance metric, the complexity of the minimum distance model is lower bounded by the complexity of self-directed learning (a topic that has recently re-gained attention in the learning theory community). To the best of our knowledge, there are few known concept classes where self-directed learning outperforms active learning in terms of sample complexity, with the notable exception of the result by Diakonikolas et al. (2023) for halfspaces over the uniform distribution. Thus, any upper bounds from our idealized setup carry over directly to self-directed learning.
> - As we highlighted in the paper, one of the most prominent applications of our model is counterexample-guided program synthesis, where it is in many cases guaranteed that the learner has perfect knowledge of the oracle and the target concept lies in the concept class.
>
> # Individual Response
> Although the sample complexity can change, contrastive examples are still very helpful in case of a  mismatch between the learner and the oracle choosing the contrastive example. Let us illustrate this with the class of halfspaces.
>
> (a) Even if the learner picks $ x_1$ at random, a single contrastive example returned by the minimum-distance oracle (with respect to the $\ell_2$ metric) is sufficient to identify the target concept. This is because the decision boundary of the target concept is perpendicular to the vector $ x'_1 - x_1 $ at the point $ x'_1 $.
>
> (b) Now, suppose the oracle returns a point $ x''_1 $ that is at most $ \gamma $ away from the true minimum-distance counterexample. In this case, it is not hard to show that if the distance from $ x_1 $ to the target hyperplane is $ r $, then the classification error of the halfspace perpendicular to $ x''_1 - x_1 $ at $ x''_1 $ is at most
> $$
> \frac{\cos^{-1} \left( \frac{r}{r + \gamma} \right)}{\pi}.
> $$
> Thus, we can conclude that even when the initial point $ x_1 $ is chosen at random, the expected classification error remains linear in $ \gamma $, even with just a single contrastive example.
>
> There are also scenarios where receiving an instance with the exact minimum distance is not necessarily optimal. For example, when learning a halfspace in the presence of label noise, the perfect minimum-distance oracle would provide the learner with outliers that may be far away from the decision boundary. Here a softer variant of the minimum-distance model may be preferable. One such alternative is an oracle that returns the closest instance for which the majority of its k nearest neighbors have the opposite label.
>
> While a detailed analysis of a mismatch between the learner’s and the oracle’s definition of the set CS is beyond the scope of our paper, we will gladly add some discussion to this end in the camera-ready version.
>
>
> References:
>
> Diakonikolas, Ilias, et al. "Self-directed linear classification." The Thirty Sixth Annual Conference on Learning Theory. PMLR, 2023.

---

### Note · Authors · 2025-08-16

We would like to thank the referees and area chair for reviewing and discussing our work. Several reviewers have commented very positively on the flexibility of our framework, the importance of the topic, as well as on the clarity of our exposition and the originality of our contribution. Of particular importance, we believe, is that reviewers noted the potential for our paper to initiate a new line of research. The concerns raised in the reviews have hopefully been sufficiently addressed in the previous author response. Perhaps worthy of emphasis in this context is the immediate relevance of our work to ongoing research in the formal methods community, even with the strong formal assumptions made in our model. Naturally, these assumptions can also be lifted to encompass more applications. We look forward to incorporating the reviewers' feedback into a revised manuscript that we hope to share with the research community at NeurIPS.

---

### Decision · Program_Chairs · 2025-09-17

**Decision:**

Accept (poster)

**Comment:**

The paper studies active learning with "contrastive examples", where the learner queries for the label of an example, and additionally receives as feedback another nearby example of a different label (either the closest such example, or an example in a given neighborhood).  They give several examples of natural concept classes where such queries offer advantages over basic membership queries.  They also find an interesting connection to the optimal mistake bounds in self-directed learning, in that such mistake bounds always lower-bound the optimal query complexity for contrastive queries.

The reviewers are unanimously in favor of acceptance.
They all agree that the setting is well-motivated and the results are novel and interesting.  Contrastive examples have been empirically useful, and this work provides theoretical insights regarding their power.
All reviews comment that the presentation is quite clear and easy to follow.

The main critique by reviewers is the idealized nature of the setting.  The learner has precise knowledge of the neighborhood from which the opposite-labeled example will be produced (similarly for the min-distance oracle, there is no slack in this min-distance criterion), and uses this information to inform its algorithm.  However, it is understandable that a theoretical framing of the problem should need to formalize some kind of understanding of how these examples will be produced (at least at first), as is common, for instance, in prior works such as local membership queries, adversarial examples, strategic examples, etc.

Another critique (though not raised by reviewers) is that the paper does not offer a general characterization of the optimal query complexity for all concept classes (and metrics), but rather merely gives several examples where such queries can (or cannot) offer a reduction in query complexity compared to basic membership queries.  Some of these examples are merely illustrative of the framework (e.g., thresholds and rectangles), while others are for concept classes of classical interest (e.g., monotone DNFs).

As a remark to the authors: In the special case of Exact Learning, there are classical general dimensions characterizing the optimal query complexity of any concept class, for fairly general families of queries (see e.g., Balcazar, Castro, and Guijarro, JCSS 2002, "A new abstract combinatorial dimension for exact learning via queries", and Balcazar, Castro, and Guijarro, COLT 2001, "A general dimension for exact learning").  I suggest including a discussion of whether the contrastive example queries considered in this work fit into that framework, and if so, how the results presented in this paper compare to what would be obtained from these general characterizations.